# Widespread occurrence and relevance of phosphate storage in foraminifera

Nicolaas Glock[1 ✉], Julien Richirt[2], Christian Woehle[3,9], Christopher Algar[4], Maria Armstrong[4], Daniela Eichner[1], Hanna Firrincieli[1], Akiko Makabe[2], Anjaly Govindankutty Menon[1], Yoshiyuki Ishitani[2], Thomas Hackl[5], Raphaël Hubert-Huard[1], Markus Kienast[4], Yvonne Milker[1,6], André Mutzberg[7], Sha Ni[1], Satoshi Okada[2], Subhadeep Rakshit[4], Gerhard Schmiedl[1,6], Zvi Steiner[7], Akihiro Tame[8,10], Zhouling Zhang[7] & Hidetaka Nomaki[2]

Foraminifera are ubiquitous marine protists that intracellularly accumulate phosphate[1], an important macronutrient in marine ecosystems and in fertilizer potentially leaked into the ocean. Intracellular phosphate concentrations can be 100–1,000 times higher than in the surrounding water[1]. Here we show that phosphate storage in foraminifera is widespread, from tidal flats to the deep sea. The total amount of intracellular phosphate stored in the benthic foraminifer *Ammonia confertitesta* in the Wadden Sea during a bloom is as high as around 5% of the annual consumption of phosphorus (P) fertilizer in Germany. Budget calculations for the Southern North Sea and the Peruvian Oxygen Minimum Zone indicate that benthic foraminifera may buffer riverine P runoff for approximately 37 days at the Southern North Sea and for about 21 days at the Peruvian margin. This indicates that these organisms are probably relevant for marine P cycling—they potentially buffer anthropogenic eutrophication in coastal environments. Phosphate is stored as polyphosphate in cell organelles that are potentially acidocalcisomes. Their metabolic functions can range from regulation of osmotic pressure and intracellular pH to calcium and energy storage. In addition, storage of energetic P compounds, such as creatine phosphate and polyphosphate, is probably an adaptation of foraminifera to $O_2$ depletion.

Phosphorus (P) is an important constituent of nucleic acids (DNA, RNA), phospholipids, phosphoproteins and adenosine triphosphate (ATP), which is responsible for the biological transmission of chemical energy[2–4]. Thus, P is essential for all living organisms on Earth. The main form of dissolved inorganic P in the ocean is phosphate, which is an important macronutrient[5] and widely used fertilizer that can cause eutrophication and deoxygenation in coastal ecosystems[6]. One recent study showed that benthic foraminifera from the Peruvian Oxygen Minimum Zone (OMZ) accumulate large amounts of phosphate within their cells[1]. Their intracellular phosphate concentrations can exceed concentrations in the surrounding seawater by 100–1,000-fold, and their high intracellular phosphate storage is probably involved in the genesis of phosphorites in the Peruvian OMZ[1].

Benthic foraminifera are ubiquitous marine protists[7]. Several foraminiferal species can inhabit extreme $O_2$-depleted environments due to their specific metabolic and morphologic adaptations[8]. These species might benefit from the current ongoing threat of ocean deoxygenation[9]. A well-studied and widespread adaptation of foraminifera to $O_2$ depletion is the intracellular storage of nitrate for denitrification, which places them as a key element within the marine nitrogen (N) cycle[10,11].

Foraminiferal denitrification is partly eukaryotic[12–14], partly complemented by bacterial symbionts[15] and, for some species, is the preferred metabolic pathway over $O_2$ respiration[16]. Nevertheless, not all foraminifera that inhabit $O_2$-depleted environments can denitrify[8] or have other known adaptation mechanisms such as fermentation[13,14].

Accumulation of P is also speculated as a common and important foraminiferal adaptation mechanism to $O_2$-depleted environments[1]. However, the regional distribution and metabolic functions of phosphate storage and subsequent use in foraminifera are poorly understood[1]. This study aims to analyse the occurrence of foraminiferal phosphate storage from diverse marine environments, and the metabolic functions of the unusually high intracellular phosphate reservoir. Finally, the importance of foraminifera for marine P cycling is discussed.

Intracellular phosphate content was measured in different foraminiferal species from diverse marine environments: an intertidal mudflat at the German Wadden Sea (Friedrichskoog), a seasonal hypoxic fjord basin (Bedford Basin), the seasonally hypoxic Sagami Bay (Japan) and the Mid-Atlantic Ridge (MAR). All analysed benthic foraminifera, except *Chilostomella ovoidea* from Sagami Bay and some species from the MAR, showed elevated intracellular phosphate levels (Fig. 1, Table 1 and

[1]Institute for Geology, University of Hamburg, Hamburg, Germany. [2]SUGAR, X-star, Japan Agency for Marine-Earth Science and Technology (JAMSTEC), Yokosuka, Japan. [3]Institute of Microbiology, Kiel University, Kiel, Germany. [4]Department of Oceanography, Dalhousie University, Halifax, Nova Scotia, Canada. [5]Institute for Chemistry, Universität Hamburg, Hamburg, Germany. [6]Center for Earth System Research and Sustainability, Universität Hamburg, Hamburg, Germany. [7]GEOMAR Helmholtz Centre for Ocean Research Kiel, Kiel, Germany. [8]Marine Works Japan Ltd, Yokosuka, Japan. [9]Present address: Miltenyi Biotec B.V. & Co. KG, Bergisch Gladbach, Germany. [10]Present address: Faculty of Medical Sciences, Life Science Research Laboratory, University of Fukui, Fukui, Japan. ✉e-mail: nicolaas.glock@uni-hamburg.de

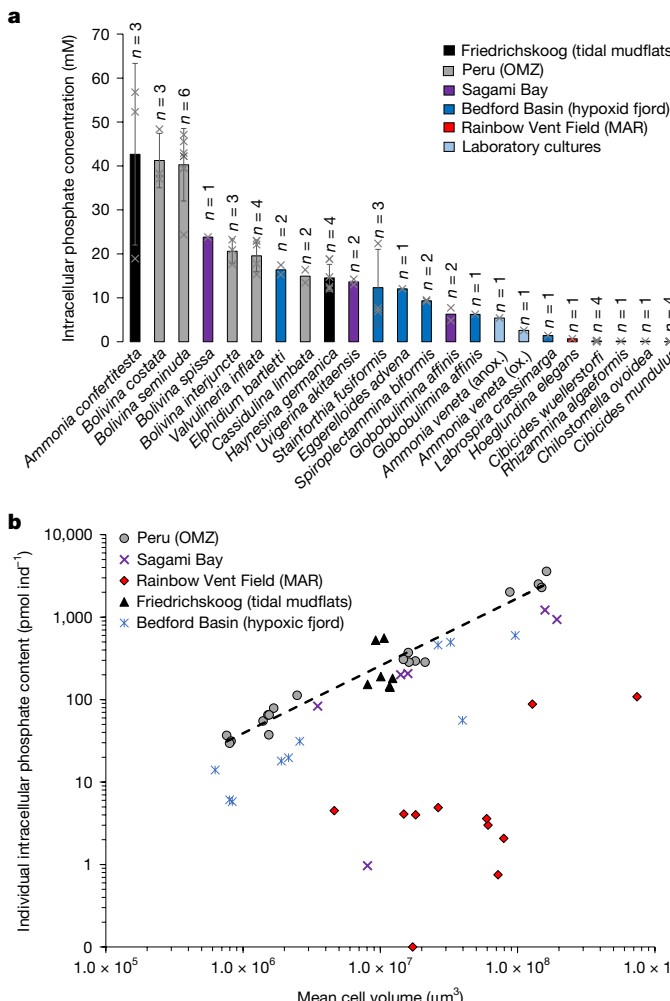

**Fig. 1 | Intracellular phosphate storage in benthic foraminifera from diverse marine environments. a**, Species sorted by mean intracellular phosphate concentrations. Column colour indicates the sampling region; *n* represents the number of biologically independent samples measured for each species. Each sample contained between one and 75 individuals, depending on the size of the species (Supplementary Table ST2). Grey crosses indicate data for individual samples. Error bars are 1 s.d. **b**, Log–log plot of individual intracellular phosphate content versus mean cell volume of the sample. Each data point represents one sample. Depending on the average body size of the species, each sample contained between one and 75 individuals (Supplementary Table ST2). Black dashed line indicates the correlation found for foraminifera from the Peruvian OMZ[1]. Anox., anoxic incubation; ox., oxic incubation.

Supplementary Table ST4), indicating the widespread occurrence of foraminiferal phosphate storage. The highest intracellular phosphate concentration was found in the species *Ammonia confertitesta* from the German Wadden Sea.

For assessment of the metabolic function of phosphate storage, intracellular P distribution was mapped in two species from Japan (*Ammonia veneta* from a cultured strain[17] and *Bolivina spissa* from bathyal Sagami Bay), using energy-dispersive X-ray spectroscopy (EDS) under scanning electron microscopy operated at cryogenic temperature (cryo-SEM). Phosphorus was accumulated, together with calcium (Ca), in cellular structures of size 0.5–2.0 µm (Figs. 2 and 3 and Extended Data Figs. 1 and 2), potential acidocalcisomes based on their elemental compositions, size and appearance. In other organisms, acidocalcisomes are usually also enriched in P and Ca, sometimes containing polyphosphate gels and granules of spherical shape[18]. The size of acidocalcisomes varies depending on the organism. The typical diameter for protists

is 0.4−0.6 µm (ref. 18), but diameters over 1 µm are not uncommon[19]. Furthermore, P compounds extracted from *A. confertitesta* were characterized, using [31]P nuclear magnetic resonance (NMR) spectroscopy. The data indicate that a large proportion of the extracted phosphate most probably originates from ATP, pyrophosphate and other polyphosphates (Supplementary Note SN1 and Supplementary Figs. SF1 and SF2). The presence of enzymes in polyphosphate metabolism for some benthic foraminifera has previously been shown[1]. In addition, due to the recent finding of a creatine phosphate metabolism in foraminifera from anoxic Namibian shelf sediments[13], the published genomes and transcriptomes of *A. confertitesta*, *A. veneta*, *Globobulimina pacifica* and *Reticulomyxa filosa* have been screened for homologues of creatine kinase, which was present in all cases (Supplementary Fig. SF4). Creatine phosphate and polyphosphates are both highly energetic, and can be used as an energy source under electron acceptor depletion[20,21]. Thus, elevated phosphate storage in foraminifera from $O_2$-depleted habitats is probably another adaptation to $O_2$ depletion. In addition, the ubiquitous occurrence of phosphate storage and accumulation of P in organelles that are possibly acidocalcisomes indicates diverse metabolic functions of this intracellular phosphate storage (Supplementary Fig. SF3).

## Widespread foraminiferal phosphate storage

These newly acquired data regarding intracellular phosphate storage in benthic foraminifera (Fig. 1 and Table 1) cover a wide range of environments, from shallow marine habitats (for example, the tidal mudflat off Friedrichskoog) to the deep sea (Rainbow Vent Field at MAR). Redox conditions vary from well oxygenated (MAR) to permanently anoxic (Peruvian OMZ centre; Supplementary Table ST1).

Elevated intracellular phosphate content has been found in species from all sampling regions. Only *C. ovoidea* from Sagami Bay and some species from the MAR did not show elevated intracellular phosphate accumulations. Species from the MAR had very low phosphate storage in general, possibly linked to well-oxygenated environments. Elevated phosphate content at the MAR was found only in *Hoeglundina elegans* and *Rhizammina algaeformis* (Table 1). Nevertheless, these two species are relatively large and, normalized to cell volume, intracellular concentrations are still very low (Fig. 1a and Table 1). Concentrations at the Peruvian OMZ, Sagami Bay and the tidal mudflats of Friedrichskoog were comparable, with a slight tendency to higher concentrations at the Peruvian OMZ, where bottom-water $O_2$ concentrations were lowest (Fig. 1 and Supplementary Table ST1). The tendency for higher phosphate storage in more $O_2$-depleted environments supports the premise that elevated intracellular phosphate content is probably an adaptation to $O_2$ depletion.

Specimens of *A. confertitesta*, having the highest intracellular phosphate concentrations measured, were sampled at the intertidal mudflats of Friedrichskoog. Oxygen penetration depth in intertidal mudflats, which are usually rich in organic matter, can be as shallow as 1 mm (ref. 22). Species of the genus *Ammonia* cannot denitrify[8], but can actively feed and thrive down to a sediment depth of several centimetres[23]. In addition, to date, no species of the genera *Elphidium* and *Haynesina*, which are typically found together with *Ammonia* spp. in $O_2$-depleted sediment, have been found to denitrify[8]. This suggests that these taxa must have alternative adaptation mechanism(s) available to cope with $O_2$ depletion. Compared with *A. confertitesta*, both *Elphidium bartletti* and *Haynesina germanica* have moderately high intracellular phosphate concentrations. Whereas kleptoplasty might be a survival strategy under $O_2$ depletion in the case of *Elphidium* and *Haynesina*[8,24], sequestered chloroplasts in *Ammonia* spp. are usually dysfunctional[25].

Thus, *Ammonia* spp. must have an alternative survival strategy under anoxic conditions. *A. confertitesta* is a putative invasive species in shallow European shelf sediments, replacing indigenous *Ammonia* spp. in

**Table 1 | Intracellular phosphate storage and mean cell volumes for benthic foraminifera from diverse marine environments**

| Species | Mean cell volume (l) | s.d. | Individual phosphate content (pmol ind$^{-1}$) | s.d. | Intracellular phosphate concentration (mM) | s.d. | Region |
|---|---|---|---|---|---|---|---|
| A. confertitesta | 9.32×10$^{-9}$ | 1.29×10$^{-9}$ | 412 | 225 | 42.7 | 20.7 | Friedrichskoog |
| A. veneta (ox.) | 3.63×10$^{-9}$ | | 20 | | 2.6 | | Culture |
| A. veneta (anox.) | 3.72×10$^{-9}$ | | 10 | | 5.4 | | Culture |
| B. costata | 7.92×10$^{-10}$ | 2.99×10$^{-11}$ | 33 | 4 | 41.2 | 6.2 | Peru (OMZ) |
| Bolivina interjuncta | 1.56×10$^{-8}$ | 7.80×10$^{-10}$ | 321 | 45 | 20.6 | 2.8 | Peru (OMZ) |
| B. seminuda | 1.69×10$^{-9}$ | 3.94×10$^{-10}$ | 69 | 25 | 40.3 | 8.3 | Peru (OMZ) |
| B. spissa | 3.49×10$^{-9}$ | | 83 | | 23.8 | | Sagami Bay |
| Cassidulina limbata | 1.96×10$^{-8}$ | 2.26×10$^{-9}$ | 290 | 7 | 14.9 | 2.1 | Peru (OMZ) |
| C. ovoidea | 8.06×10$^{-9}$ | | 1 | | 0.1 | | Sagami Bay |
| Cibiscides mundulus | 6.79×10$^{-8}$ | 9.46×10$^{-9}$ | 2 | 1 | 0 | 0 | Rainbow Vent Field (MAR) |
| Cibicidoides wuellerstorfi | 1.91×10$^{-8}$ | 5.04×10$^{-9}$ | 3 | 2 | 0.2 | 0.1 | Rainbow Vent Field (MAR) |
| Eggerella advena | 2.59×10$^{-9}$ | | 31 | | 12.1 | | Bedford Basin |
| E. bartletti | 2.94×10$^{-8}$ | 4.37×10$^{-9}$ | 478 | 27 | 16.4 | 1.5 | Bedford Basin |
| Globobulimina affinis | 1.76×10$^{-7}$ | 2.48×10$^{-8}$ | 1,078 | 201 | 6.3 | 2.0 | Sagami Bay |
| Globobulimina affinis | 9.62×10$^{-8}$ | | 600 | | 6.2 | | Bedford Basin |
| H. germanica | 1.15×10$^{-8}$ | 9.49×10$^{-10}$ | 165 | 25 | 14.5 | 3.1 | Friedrichskoog |
| H. elegans | 1.28×10$^{-7}$ | | 88 | | 0.7 | | Rainbow Vent Field (MAR) |
| Labrospira crassimarga | 3.97×10$^{-8}$ | | 56 | | 1.4 | | Bedford Basin |
| R. algaeformis | 7.40×10$^{-7}$ | | 108 | | 0.1 | | Rainbow Vent Field (MAR) |
| Spiroplectammina biformis | 2.02×10$^{-9}$ | 1.74×10$^{-10}$ | 19 | 1 | 9.3 | 0.2 | Bedford Basin |
| Stainforthia fusiformis | 7.52×10$^{-10}$ | 1.11×10$^{-10}$ | 9 | 5 | 12.3 | 8.7 | Bedford Basin |
| Uvigerina akitaensis | 1.50×10$^{-8}$ | 1.28×10$^{-9}$ | 203 | 5 | 13.6 | 0.9 | Sagami Bay |
| Valvulineria inflata | 1.35×10$^{-7}$ | 3.28×10$^{-8}$ | 2,603 | 684 | 19.5 | 3.6 | Peru (OMZ) |

Data for the Peruvian OMZ are cited from ref. 1.

these environments, and probably originated from Asia[26,27]. Thus, it is possible that the high intracellular phosphate storage of *A. confertitesta* could provide a competition advantage over other autochthonous species in these environments. *Bolivina seminuda* and *Bolivina costata* from the Peruvian OMZ showed high phosphate storage, comparable to that of *A. confertitesta*. These *Bolivina* species can dominate anoxic and even sulfidic habitats[28,29], and are able to denitrify[11,16]. This suggests that the ability to denitrify and high phosphate storage, as adaptations to O$_2$ depletion, are not mutually exclusive, and provide even further competitive advantages for thriving in environments in which oxygen is low or absent.

## Diverse metabolic functions

Newly acquired coupled cryo-SEM–EDS data acquired for *A. veneta* and *B. spissa* indicate that phosphate storage is related to additional metabolic functions other than being an adaptation to O$_2$-depleted conditions. Analyses on cells of *B. spissa* indicate that P is accumulated with Ca, and sometimes with Mg, in solid, granule-like structures (Fig. 2 and Extended Data Fig. 2) of appearance similar to other Ca polyphosphates[30,31]. In *A. veneta*, P is accumulated in round structures 0.5–2.0 μm in diameter (Fig. 3). Phosphorus seems to be less locally concentrated in *A. veneta* than in *B. spissa*, and not in the form of solid Ca polyphosphates, which indicates a generally lower intracellular phosphate content in *A. veneta* compared with *B. spissa*. This is supported by the intracellular phosphate concentrations that have been measured for those species, which were also lower for *A. veneta* (Fig. 1 and Table 1). These round structures, which are abundant within cells of *A. veneta* and that have been imaged with cryo-SEM (Fig. 3a,b and Extended Data Fig. 1), are similar in size and shape to acidocalcisomes[19], organelles known to accumulate pyrophosphate (diphosphate), and

to granules that are enriched in Ca and polyphosphates and other metals such as Mg. Nevertheless, we cannot exclude the possibility that some of these structures are autophagosomes that can also accumulate polyphosphates[19]. Autophagosomes have a similar size and shape, although they usually show the presence of membranous debris and have a more irregular shape[19]. Digestive food vacuoles known from foraminifera have a completely different size and shape[32] and can most probably be excluded. Putative acidocalcisomes are either absent or empty in transmission electron microscope (TEM) images on thin sections of *A. veneta*, because abundant and empty round vesicles of the same size are visible (Fig. 3c,d and Extended Data Figs. 3 and 4). During conventional preparation methods of thin sections for TEM imaging, cell fixation involves a dehydration step, decreasing the water content and resulting in the loss of soluble content within the fixed specimen[33]. In addition, no P accumulations could be found using coupled TEM–EDS on the same thin sections, except in one vesicle that was still filled (Fig. 3d and Extended Data Figs. 3 and 4). When acidocalcisomes are observed with TEM, cryofixation is generally used rather than the traditional thin-section preparation for TEM[19]. Nevertheless, it has already been recorded that, even when using cryo-TEM, it can be complicated to distinguish acidocalcisomes from other vesicles and vacuoles[19]. The presence in TEM observations of many empty vesicles of size and shape similar to the round, P-rich structures seen on cryo-SEM–EDS images, gives rise to speculation that many of the empty vesicles observed in *A. veneta* using TEM are the same structures as the filled ones on cryo-SEM observations, and may possibly be acidocalcisomes.

Acidocalcisomes, which are common in prokaryotes and eukaryotes[34] and abundant in many protists[19] have never previously been described in foraminifera, despite a long history of ultrastructural analyses of foraminiferal cells[32]. This is probably on account of the artefacts seen

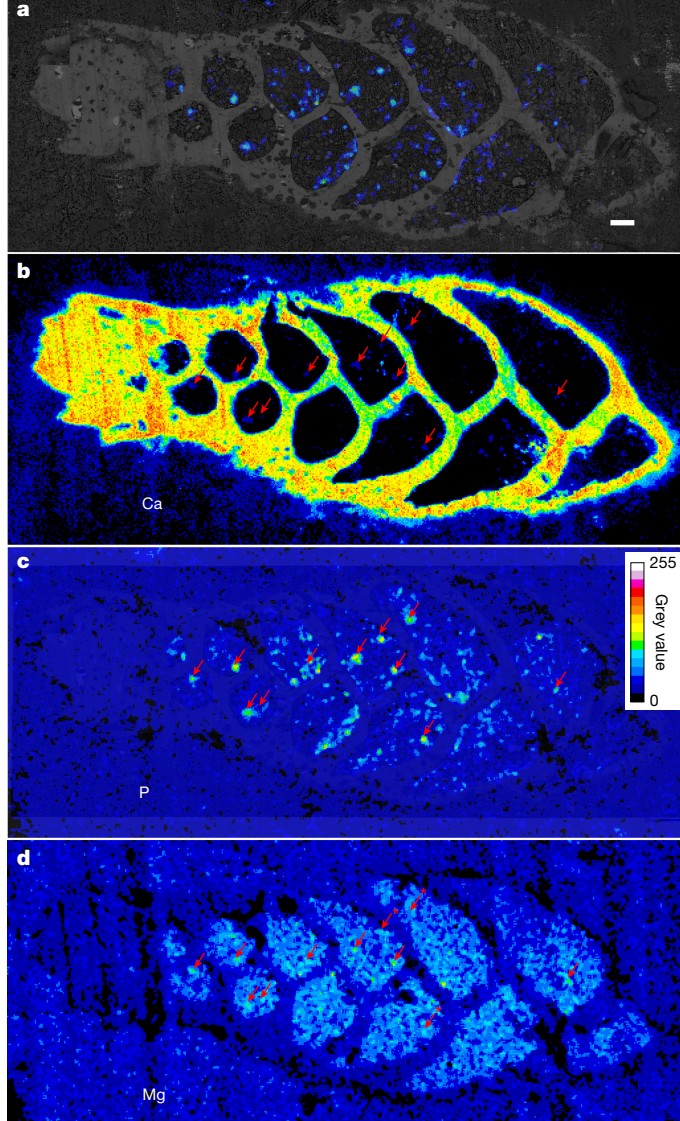

**Fig. 2 | Elemental maps of *B. spissa* cross-section. a–d**, Cryo-EDS mappings of P and Ca and cryo-SEM images on cross-section of a cryo-fractured specimen of *B. spissa*. **a**, P distribution, mapped with cryo-EDS, overlain by the cryo-SEM image of the same region. **b–d**, Overview of the distribution of P (**b**), Ca (**c**) and Mg (**d**). Arrows indicate structures enriched in both Ca and P, and often in Mg. *, Structures enriched in P and Ca but not in Mg. The cryo-SEM–EDS experiment was repeated on three different specimens of *B. spissa*. Scale bar, 20 μm.

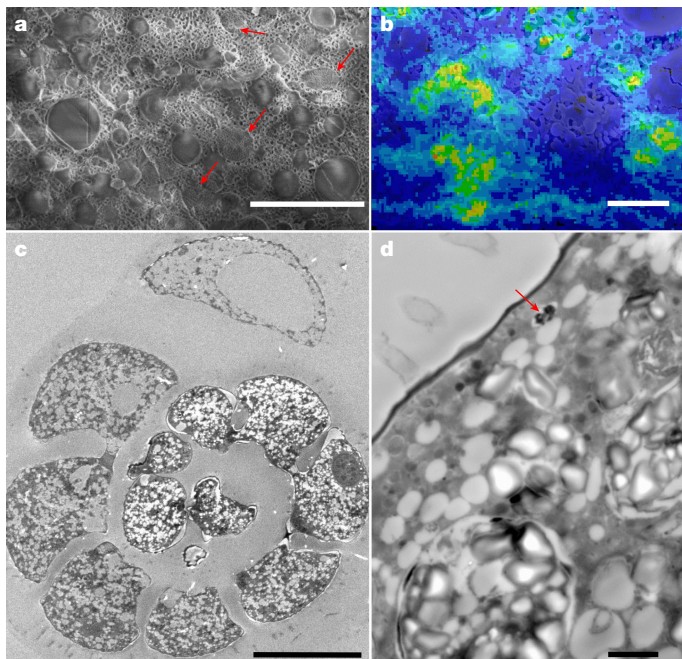

**Fig. 3 | Ultrastructure of *A. veneta* specimens. a**, Cryo-SEM image (secondary electron mode) of cross-section of a cryo-fractured specimen of *A. veneta*. Note the abundant circular structures (denoted by red arrows) that are possibly acidocalcisomes. **b**, P distribution, mapped with cryo-EDS, is shown in green on a cryo-fractured specimen of *A. veneta*, overlain by the cryo-SEM image of the same region (SEM image and P distribution are shown individually in Supplementary Fig. SF2). **c,d**, The circular, slightly P-enriched structures are absent on TEM images of thin sections of *A. veneta*. Instead, there are abundant empty vesicles visible on the thin sections, which indicates that these structures might have lost their content during fixation and subsequent embedding and polymerization processes. Only one of these vesicles has been found that was not empty (marked with the red arrow in **d**). This structure was the only structure with measurable P content using coupled TEM–EDS (Extended Data Fig. 4). The cryo-SEM–EDS experiment was repeated on eight different specimens of *A. veneta*, and the TEM–EDS experiment on three different specimens. Scale bars, 5 μm (**a**), 2 μm (**b,d**) and 50 μm (**c**).

during the non-cryogenic preparation of thin TEM sections described above. Acidocalcisomes can have diverse functions, including the storage of polyphosphate and pyrophosphate and participation in the related energy metabolism, intracellular pH and osmoregulation and Ca storage for the conservation of Ca homeostasis[18]. If the observed structures are acidocalcisomes, or have a similar function, this would have several implications that could explain why high phosphate storage might provide crucial advantages for species that can accumulate more phosphate than others, including *B. costata* and *A. confertitesta* (Fig. 1), as shown below.

(1) Rapid osmoregulation. Rapid hydrolysis or synthesis of polyphosphates in acidocalcisomes, which increases or decreases the intracellular electrolyte concentration, has been shown as a reaction to hypo- or hyperosmotic stress in *Trypanosoma cruzi*, a protist belonging to Excavata[35]. Both *A. confertitesta* and *B. costata* are often found in shallow marine environments that are strongly influenced by tidal cycles[26,27,29], and, thus, undergo marked salinity changes, which suggests that this mechanism might be advantageous for rapid osmoregulation.

(2) Rapid generation of metabolic energy. Because *A. confertitesta* and *B. costata*, as well as other benthic foraminiferal species, are often exposed to anoxia, the storage of energy-rich polyphosphates most probably provides a source of rapidly available energy when these organisms run out of terminal electron acceptors, such as $O_2$ or nitrate. Due to their energy-rich P bonds, polyphosphates are a powerful energy source[30], and sulfur bacteria from $O_2$-depleted environments have previously been shown to utilize polyphosphates as an energy source when depleted of of electron acceptors[36]. Previous comparative genomics analyses have indicated the presence of a polyphosphate metabolism in multiple species of foraminifera[1].

(3) Intracellular pH regulation with possible relevance for calcification. Most of the foraminifera analysed in this study have calcareous tests. The foraminiferal calcification mechanism involves the elevated concentration of carbonate ions in vacuoles with elevated pH[37–39]. Acidification of acidocalcisomes is driven by vacuolar proton pumps[18] and elevates pH in other cell compartments that might be used in the foraminiferal calcification pathway. Thus, if the organelles in foraminifera described above are indeed acidocalcisomes, they might play an important role in the foraminiferal calcification

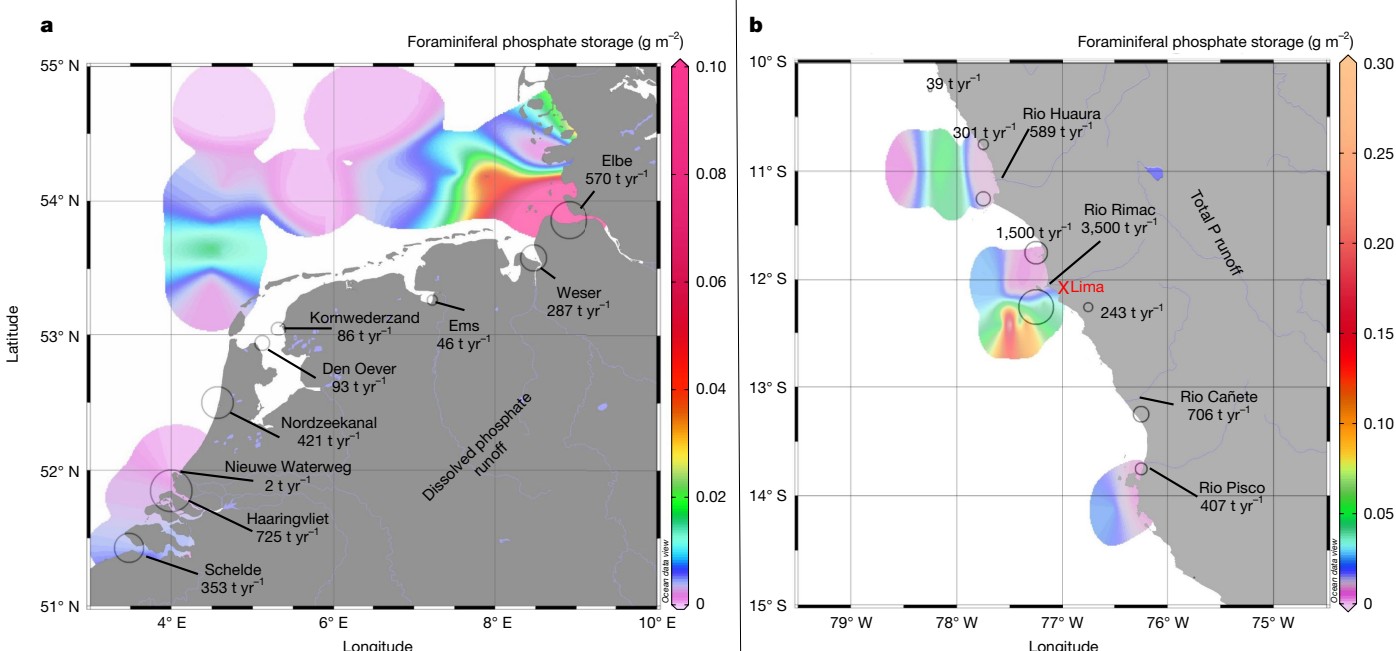

**Fig. 4 | Distribution of benthic foraminiferal storage. a,b,** Maps showing the distribution of benthic foraminiferal phosphate storage in the Southern North Sea (**a**) and off Peru (**b**). Benthic foraminiferal phosphate storage was estimated using the composition of living benthic foraminiferal assemblages from 135 stations in the Southern North Sea[51–57] and 35 off Peru[1,29] (for details of calculations, see Methods). Maps show the weighted mean. Circles and numbers indicate regional riverine phosphate[58] runoff (**a**) and total P (ref. 59) runoff (**b**). More detailed budget calculations show that benthic foraminifera might buffer approximately 37 days of riverine phosphate runoff to the Southern North Sea, and roughly 21 days of riverine total P runoff to the Peruvian coast between 10° S and 15° S (Supplementary Note SN8). Maps were generated using Ocean Data View software[60].

mechanism, with high intracellular phosphate storage probably providing advantages in regard to intracellular pH regulation for calcification. Previous studies show that pH regulation in acidocalcisomes can be crucial in some organisms to adapt to changes in environmental pH[40]. There are often strong pH gradients in pore waters of tidal flat sediments, within a few millimetres[41]. Thus, species such as *A. confertitesta* often undergo rapid pH changes in their microenvironment, and high phosphate storage is probably advantageous as an adaptation to such changes.

Despite the relevance of phosphate storage in acidocalcisomes, the metatranscriptome of foraminifera from a natural anoxic sediment environment off Namibia showed that they encode genes for a creatine phosphate metabolism[13]. New comparative genomics and transcriptomics analyses on the species *A. confertitesta*, *A. veneta*, *G. pacifica* and *R. filosa* show that these species also possess and/or express the enzyme creatine kinase and, thus, are able to synthesize and metabolize creatine phosphate (Supplementary Note SN4 and Supplementary Fig. SF4). Creatine phosphate is highly energetic, can rapidly regenerate ATP from ADP and can provide homeostasis of cellular bioenergetics[42]. Especially in muscle cells, creatine phosphate can rapidly provide energy during periods of elevated activity, or even bursts with high energetic demand[43], and is an important contributor at the beginning of anaerobic muscle metabolism[44]. Thus, it has previously been suggested that creatine phosphate metabolism, observed in foraminifera from anoxic sediments at the Namibian Shelf[13], provides energy storage for sudden energetic bursts[13], such as feeding by phagocytosis[13]—that is, vacuolar ingestion of food particles and prey[13]. Because creatine kinase is encoded in the genome of *A. confertitesta* (Supplementary Note SN4 and Supplementary Figs. SF4 and SF5), creatine phosphate storage probably confers on *A. confertitesta* the possibility of staying active even when exposed to anoxia. Creatine phosphate might even be utilized for energy bursts that are required for the predatory activity of this species that has been observed even under anaerobic conditions several centimetres below the sediment–water interface[23]. Thus, polyphosphates within acidicalcisomes and creatine phosphate are most probably additional adaptations of certain foraminiferal species to oxygen depletion. Although the metabolic functions of intracellular phosphate storage in foraminifera might be quite diverse (for example, energy supply under $O_2$ depletion, regulation of intracellular pH and electrolyte concentration), it is easy to explain why deep-sea foraminifera from the MAR show lower intracellular phosphate accumulations. Except in some extreme examples, environmental conditions in the well-oxygenated deep sea are very stable, and organisms in this environment rarely experience severe $O_2$ depletion or marked fluctuations in pH or salinity.

Finally, initial exploratory [31]P-NMR data on P compounds extracted from *A. confertitesta* confirm the presence of high levels of pyrophosphate (Fig. 1 and Supplementary Note SN1). In addition, ATP was present in high concentrations and smaller amounts of inorganic polyphosphates were probably extracted, too, although these are overlain by ATP peaks (Fig. 1 and Supplementary Note SN1). One peak in the [31]P-NMR spectrum is close to that of creatine phosphate, but slightly drifted. The [31]P-NMR spectrum measured directly on living specimens of *A. confertitesta* is much more complex and indicates a mixture of more diverse P compounds. Thus, the presence of creatine phosphate in the extracted P compounds should be confirmed in future studies.

## Relevance for P cycling and the environment

It is well documented that benthic foraminifera performing denitrification are important for the marine nitrogen cycle, and even outcompete denitrifying bacteria in some $O_2$-depleted environments due to their high abundance[8,11,16,45]. One recent study showed that the intracellular phosphate concentration in foraminifera from the Peruvian OMZ can be 100–1,000 times higher than in the surrounding

pore waters, making them an important and previously overlooked reservoir for benthic inorganic P (ref. 1). That study also indicates that high intracellular phosphate content facilitates the nucleation of phosphorites in this environment[1], in a way similar to how certain sulfur bacteria initiate phosphogenesis in regard to their high accumulation of polyphosphates[36,46]. The parallel enrichment of polyphosphates and Ca in the putative acidocalcisomes (Fig. 2) supports this hypothesis, because phosphorites are phosphatic rocks consisting, to a large degree, of apatite[47,48]. Rapid hydrolysis of polyphosphates post mortem might initiate apatite nucleation in favourable environments.

A rough extrapolation based on *A. confertitesta* density in Wadden Sea sediment during a bloom, and on individual intracellular phosphate content, suggests that the phosphate reservoir in *A. confertitesta* is huge (Supplementary Note SN6). The population density of living *A. confertitesta* during the bloom at Friedrichskoog was 417 individuals (ind) cm$^{-3}$ (Supplementary Note SN6), and individual phosphate storage was 413 pmol ind$^{-1}$. The Wadden Sea has an extension of 11,500 km$^2$, resulting in a total amount of roughly 1,880 t phosphate stored in *A. confertitesta* within the top 1 cm of sediment. This represents about 5% of the yearly consumption of P-containing fertilizer in Germany in 2021/2022 (Supplementary Note SN6) within only a single foraminiferal species during the snapshot of a bloom, which provides a rough estimate of the importance of the entire group of foraminifera for oceanic P cycling.

A more detailed analysis of benthic foraminiferal phosphate storage in sediments from 135 stations in the Southern North Sea (Supplementary Table ST6) and 35 at the Peruvian Margin[28] (Supplementary Table ST7) shows that benthic foraminifera might be an effective buffer for riverine phosphorus runoff in these regions (Fig. 4, Supplementary Note SN8 and Supplementary Fig. SF6). Total foraminiferal phosphate storage in the region of interest of the Southern North Sea (0.0059 ± 0.0014 g m$^{-2}$; 1 standard error of the mean (s.e.)) is equivalent to roughly 37 days of riverine phosphate runoff in this region (2,583 t yr$^{-1}$). Foraminiferal assemblages between 10° S and 15° S off Peru store 0.0315 ± 0.0101 g m$^{-2}$ (1 s.e.), and thus may buffer approximately 21 days of riverine total P runoff in this region. Note that, because data for total riverine phosphate runoff from this region were not available, budgets had to be calculated slightly differently by using, instead, riverine total P runoff (Supplementary Note SN8).

At the Peruvian OMZ, foraminiferal phosphate storage may be of particular importance. Phosphate from remineralized organic matter adsorbs to iron oxide layers under oxic conditions and is effectively trapped in oxic sediments, whereas it escapes to the water column and is efficiently recycled under markedly O$_2$-depleted conditions[49]. The high abundance of benthic foraminifera in this region might dampen this phosphate recycling, reduce productivity and thus act as a negative feedback mechanism in the expansion of the OMZ that has been observed since the 1960s[50].

Finally, these results indicate that benthic foraminifera can be an important buffer to counteract eutrophication. Denitrifying foraminifera already mitigate eutrophication by reducing reactive nitrate to non-reactive N$_2$ gas. These new findings regarding phosphate storage in foraminifera indicate that these organisms also contribute to de-eutrophication related to their high phosphate accumulations, making them key mediators that probably buffer the effects of coastal eutrophication. Budget calculations of total phosphate stored in benthic foraminifera in the Southern North Sea and the Peruvian OMZ indicate that this phenomenon is also globally important. Nevertheless, reactive P stored in foraminifera is not completely removed from the environment, such as nitrate during denitrification. Thus, intracellular foraminiferal phosphate storage can be seen more as temporary withdrawal from the system—that is, a 'hidden' standing stock that is temporarily unavailable for primary production.

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

## Methods

### Sediment sampling

For sampling of living benthic foraminifera, samples were taken during various surveys at Sagami Bay (Japan), the Rainbow Vent Field at the MAR, the Bedford Basin (Canada) and the German Wadden Sea (Friedrichskoog). Samples at Sagami Bay were taken during a RV *Kaimei* cruise in September 2019 and on RV *Yokosuka* cruises in October 2022 and May 2023. Sediment was retrieved using a multicorer during the former and a push core during the latter at stations in central Sagamy Bay (NSB site; Supplementary Table ST2). Samples from the region around the Rainbow Vent Field were retrieved during RV *Meteor* cruise no. M176/2 in September 2021, using a multicorer at six stations (Supplementary Table ST2). Samples from the Bedford Basin were retrieved in March 2022 on board the dive vessel *EastCom*, using a multicorer at three stations (Supplementary Table ST2). Sediments from the intertidal mudflats in Friedrichskoog were retrieved manually at one station in November 2021 and May 2023. The top 1 cm of sediment was scraped off by spoon. Surface sediments from the brackish water salt marsh of Hirakata Bay, Yokohama (Japan) were collected in 2015 (Supplementary Table ST2), and isolated *A. veneta* strains were maintained in the laboratory.

### Preparation of living foraminifera for intracellular phosphate analyses

The top 1 cm of sediment was directly wet sieved over a 125 or 250-μm mesh within a time frame of 2 h following core retrieval, using either filtered regional surface water or nitrate- and phosphate-free artificial seawater (ASW) at ambient salinity prepared using Red Sea Salt. Only certain samples from the Bedford Basin were sieved within 2 days following sampling, using filtered seawater from the Bedford Basin. Sediments retrieved from Friedrichskoog in March 2023 were directly sieved in the field, using surface water.

Living foraminifera specimens for intracellular phosphate analyses were wet picked from the coarse residue (over 250 μm for Friedrichskoog samples and over 125 μm for the remainder). In total, 34 samples were picked that included 15 different benthic foraminifera species. Specimens of *A. veneta* were obtained from the clonal strain cultured at the Japan Agency for Marine-Earth Science and Technology (JAMSTEC). For a description of culturing methods, see below. One *A. veneta* sample was prepared from oxic incubation and one from anoxic. Each foraminiferal sample contained between one and 75 living foraminifera specimens depending on the average size of the species (Supplementary Table ST2). Note that the sample of *R. algaeformis* contained only one large fragment of the organism, which was visibly filled with cytoplasm. All samples were photographed with a stereomicroscopic camera for subsequent determination of foraminiferal biovolume. Afterwards, foraminifera were cleaned and phosphate extracted using the methods described in ref. 1. Specimens were rinsed with phosphate-free ASW, prepared from Red Sea Salt, then transferred to centrifuge tubes with the lowest amount of ASW possible. Next, 3 ml of reverse osmosis water (with conductivity of 0.055 μS cm$^{-1}$) was added to samples. Within the water, foraminiferal specimens were broken up using a clean pipette tip. A procedural blank underwent the same procedure without foraminiferal specimens, for blank corrections of $NO_3^-$ analyses (13 procedural blanks in total). All samples were frozen at −20 °C for at least 2 h and subsequently thawed. This procedure was repeated three times. Freeze–thaw injuries damage plasma membranes and increase their permeability[61,62]. Subsequently, samples were filtered through sterile 0.2-μm cellulose acetate filters.

A slightly adapted protocol was used for the extraction of P compounds for $^{31}$P-NMR; these samples were retrieved from Friedrichskoog in March 2023. Two replicates were picked, each containing around 1,000 living specimens of *A. confertitesta*. Specimens were rinsed with phosphate-free ASW prepared from Red Sea Salt and subsequently transferred to microcentrifuge tubes with the lowest amount of ASW possible. Next, 600 μl of heavy water (D$_2$O) with 1 M KOH was added to samples. Within the water, foraminiferal specimens were broken up using a clean pipette tip. Samples were frozen at −20 °C for at least 2 h and then thawed. This procedure was repeated three times. Subsequently, samples were centrifuged. A further sample was picked that contained roughly 1,500 specimens of *A. confertitesta*, which were transferred unharmed to a NMR tube containing D$_2$O and with a salinity of 28.

### Analyses of extracted intracellular phosphate

Filtered samples were analysed for total dissolved phosphate by segmented flow-injection analysis using a QUAATRO39 (Seal Analytical) autoanalyser, which included a XY2-autosampler unit, at GEOMAR Helmholtz Centre for Ocean Research Kiel (Kiel, Germany). Samples from the MAR were analysed, using the same methods, directly on board during RV *Meteor* cruise no. M176/2. The system set-up included four channels—for nitrate + nitrite, silicate, nitrite and phosphate—but only the phosphate data were used within this study. The method used for phosphate analysis corresponds to Q-064-05 Rev. 8 (developed by Nederlands Instituut voor Onderzoek der Zee; detection limit 0.004 μmol l$^{-1}$ and described by QuAAtro Applications).

### Biovolumetric determination of living foraminifera

Total foraminiferal cell volume of each species was estimated following previously published methods[63]. We assumed that internal test volume corresponds to 75% of total test volume and is completely filled with cytoplasm[64]. Methodology and equations used for precise biovolume estimation in several benthic foraminifera species are given in ref. 65 but, unfortunately, none of the species analysed in our investigation are listed in their study; therefore, the closest geometric shape was used for biovolume estimation (Supplementary Table ST3). In total, the biovolume of 850 foraminiferal specimens was determined.

Note that, for the two shapes related to *B. spissa* (cone with elliptic base) and *C. wuellerstorfi* (triaxial ellipsoid), the heights of the specimens would have been required, which were not visible on the images. The following approximations have been used instead: for *B. spissa* specimens, height was estimated by the previously determined mean height of the species (133 ± 7 μm)[16]; for *C. wuellerstorfi*, the average ratio of the shortest diameter on the spiral side to the height was determined (0.424 ± 0.029, 1 s.e.), using published images of the species[66] (Supplementary Table ST5).

### Determination of living abundance

One sample retrieved from Friedrichskoog in March 2023 was taken to determine the abundance and population density of *A. confertitesta*. For this sample, we used a square-shaped metal frame with a side length of 10 cm. Within this metal frame, the top 1 cm of sediment was scraped off and collected in a polyethylene bottle. Next, a mixture of ethanol and Rose Bengal (2 g l$^{-1}$) was added to the bottle until ethanol concentration exceeded 70%. The jar caps were cleaned and applied tightly, the height of the sediment in the jar was marked and the jars stored for at least 14 days at room temperature until further analysis. Subsequently, stained samples were wet sieved over a 125-μm mesh, dried at around 40 °C and the jars filled with water up to the sediment fill mark level. The volume of water representing bulk sediment volume was measured in a graduated cylinder (approximate accuracy ±5 cm$^3$). Subsequently, samples were split using a dry splitter, and specimens of *A. confertitesta* stained with vital raspberry red were counted under the microscope. Living stained foraminifera were fixed in plummer cells.

### Culture of *A. veneta*

Cultures of *A. veneta* were from the same strain used by Ishitani et al.[17] isolated first in 2015. Specimens were cultured in ASW at a salinity of 35, at 23 °C under 14/10 h light/dark cycles. The specimens used for this

experiment were fed, frozen, dead *Dunaliella salina* (no. NIES-2257). We isolated five specimens with shell diameter 150–300 μm from sub-culture into 35-mm culture dishes with 5 ml of ASW, with culture for 4 days under both oxic and anoxic conditions. We cultured normally for oxic conditions in an AnaeroPack-Anaero, which can maintain 0% $O_2$ and 15% $CO_2$ for anoxic conditions.

## Comparative genomics and metabarcoding

Creatine kinase homologues were identified with the KEGG KAAS tool[67] (species: hsa, mmu, rno, dre, dme, cel, ath, sce, ago, cal, spo, ecu, pfa, cho, ehi, eco, nme, hpy, bsu, lla, mge, mtu, syn, aae, mja, ape, mbr, ddi, tet, smin, pti, ehx, gtt, ngr, tva, tbr, spar) as K00933. The following genomes and transcriptomes were screened for comparative genomic analysis:

- the genome of *R. filosa* (downloaded from the National Center for Biotechnology Information (NCBI), accession no. GCA_000512085.1_Reti_assembly1.0)[68];
- the transcriptome of *Globobulimina* spp. (NCBI accession no. GGCD00000000.1)[12];
- the transcriptome of *Ammonia* spp. (NCBI accession no. GIDR00000000.1).

Subsequently, published transcriptome data of *Ammonia* spp. were collected from the NCBI database for further identification of this taxon at species level by metabarcoding. Collected raw reads were quality filtered with FASTX-Toolkit 0.0.13 (ref. 69), and those with fewer than 50 bases—or that included ambiguous barcodes and showed poor quality ($q$-score <20)—were removed.

## NMR spectroscopy

All NMR experiments were carried out on a Bruker 600 MHz Avance III HD spectrometer (14.09 T, 600.13 MHz for [1]H, and 242.94 MHz for [31]P) at 298 K in $D_2O$. One-dimensional [31]P{[1]H} spectra were obtained utilizing a 30° excitation pulse and relaxation delay of 1.0 s. The waltz16 sequence was implemented for proton decoupling. Spectra were acquired at a spectral width of 96,153.84 Hz and 65,536 time domain data points, by recording 1,024 scans for extracted samples and 3,584 for the sample containing live foraminiferal cells[31]. P chemical shifts were referenced to external phosphoric acid (external measurement). Data were acquired using TopSpin v.3.6.4, and all spectra were processed with Topspin v.4.1.4, applying zero filling and an exponential multiplication of the free induction decay with a line-broadening factor of 1.0 Hz.

## Cryofixation for cryo-SEM–EDS

*Bolivina spissa* specimens were isolated from the topmost 1 cm of sediment directly onboard, immediately following sampling under a stereomicroscope; *A. veneta* specimens were picked from cultures. Cryofixation followed the protocols of ref. 70. For cryofixation of foraminifera, conductive glue comprising 30 wt% graphite oxide and glycerol was used with probe sonication, following the method described in ref. 71 with solvent modification. Glycerol was applied as a cryoprotectant and viscous dispersant to prevent sinking of foraminifera deep into the glue. The glue was pasted onto an aluminium rivet (diameter 3 mm), each foraminifer specimen was mounted on the glue using an eyelash brush and was then immediately frozen in semifrozen isopentane at −159.8 °C. The rivet was then mounted on an ultramicrotome (Ultracut S equipped with FCS, Leica Microsystems, operated at −130 °C) and the cross-section faced using a diamond knife (Diatome). Faced samples were stored in a container below −160 °C until required for cryo-SEM observation.

## Cryo-SEM–EDS

In total, three *B. spissa* specimens (two from the October 2022 cruise and one from the May 2023 cruise) and eight *A. veneta* specimens were analysed. Cryo-SEM observation was performed on a Helios G4 UX

(Thermo Fisher Scientific) equipped with a cryogenic stage and a cryopreparation chamber (catalogue no. PP3010T, Quorum Technologies). EDS analyses were performed on an Octane Elite Super (C5) (AMETEK), which was attached to the cryo-SEM (software TeamEDS, v.4.6.0052.0238). The sample was mounted on a transfer shuttle in liquid nitrogen, then vacuum transferred to the cryopreparation chamber. Water was sublimed at −80 °C for 8 min to expose the organelle structure, and Cr was then coated by magnetron sputtering at 20 mA for 60 s. Note that sublimation does not melt or sublime glycerol-based glue. We selected Cr because conventional Pt or Au sputtering causes overlap in EDS signals, including P; the K-lines of N and P appear at 0.392 and 2.013 kV, respectively, and the M-lines of Pt and Au at 2.048 and 2.120 keV, respectively. By contrast, the L-line of Cr appears at 0.573 eV, overlapping only with O, and no signals appear between it and its K-line at 5.414 keV. The sputtered sample was transferred to the SEM chamber and maintained below −140 °C; cross-sectional morphology was imaged by secondary electron at 2 kV with 50 V of antibias on the sample, and EDS mapping was performed at 20 kV without antibias.

Spectral treatment aiming to deconvolve signal from noise was performed on EDS elemental maps. Conventional quantitative EDS analyses use correction by atomic number (Z), absorption (A) and X-ray fluorescence (F), called the ZAF method, assuming that the surface is completely flat and that elemental composition, along with depth direction, is homogeneous. However, our cryo-SEM and cryo-EDS maps did not meet these requirements and the apparent EDS maps were correlated with background, so that we could see the similarity between count-per-second (CPS) maps and the EDS maps of low-intensity atoms. Therefore, we tried to suppress the position-dependent background signal[72]. EDS maps are generated from the number of counts in which X-rays from atom A appear, and count $I_A$ is composed of pure signal $S_A$ and background $B_A$, where background is mostly due to the bremsstrahlung effect. Because CPS is the sum of signal from all atoms and background, $CPS = \Sigma_i S_i + B = \Sigma_i I_i + B_i$, where $i$ indicates the atom of interest, $B$ is the summation of background intensity for all of the energy and $B_i$ is the summation of background intensity at energy ranges in which characteristic X-ray peaks are absent. The background mostly derives from bremsstrahlung and is thus dependent on beam condition, and here we assume that background shape is the same in any position of EDS maps. The intensity-to-noise ratio ($R$) of atom $i$ at position (m, n) is defined as equation (1):

$$R(m, n) = I_i(m, n)/B_i(m, n) = S_i(m, n)/B_i(m, n) + 1$$
$$= I_i(m, n)/(CPS - \sum_i I_i(m, n), \tag{1}$$

which is related to conventional signal-to-noise ratio $S_i/B_i$. We used $R$ to emphasize localized minor elements hidden under the strong background.

Finally, colocalization of SEM images and the EDS elemental map was performed manually using calcium distribution maps, the main compound of the test in calcitic foraminifera. When needed, we performed EDS map scaling and/or rotation without deformation (that is, warping). For enhanced visualization of elemental distribution on EDS maps, a 16-colour look-up table was applied on EDS maps without grey value modification (Fig. 2c). Then, for all elements (excluding Ca), one-pixel median filtering was performed to smoothe the elemental distribution map and identify enriched ultrastructures regarding the element of interest. Image treatment was performed using the software Fiji[73].

## TEM–EDS

Foraminifera specimens were fixed with 2.5% glutaraldehyde in filtered ASW for at least 24 h at 4 °C. They were then embedded in 1% aqueous agarose and cut into cubes of roughly 1 mm³. Fixed specimens were embedded in 1% aqueous agarose, decalcified with 0.2% ethylene glycol-bis(2-aminoethylether)-N,N,N′,N′-tetraacetic acid in 0.81 mol l⁻¹

aqueous sucrose solution (pH 7.0) for several days and then rinsed with filtered seawater. For measurement of P in cells with EDS, we did not conduct postfixation with osmium tetroxide, which was the over-lapped energy peak position of P. Specimens embedded in agarose were rinsed with filtered ASW, stained with 2% uranyl acetate solution for 2 h at 4 °C, dehydrated in a graded ethanol series and embedded in epoxy resin (Quetol 651).

Ultrathin sections (100 nm) were cut using a diamond knife on an Ultracut S ultramicrotome and then stained with 2% aqueous uranyl acetate and lead staining solution (0.3% lead nitrate and 0.3% lead acetate, Sigma-Aldrich). TEM−EDS imaging was performed on a Tecnai G2 20 (Thermo Fisher Scientific), equipped with a bottom-mounted 2k × 2k Eagle charge-coupled device camera (Thermo Fisher Scientific) and a RTEM-S 61700ME EDS detector, (AMETEK) at an acceleration voltage of 200 kV. Note that the elements used for staining are heavier than Au and do not overlap with P.

### Calculation of total phosphate storage in living foraminiferal assemblages

The total dissolved inorganic phosphate pool stored in foraminifera in the sediment column ($\sum DIP_{i\,sed.}$, in mmol m$^{-2}$) was calculated for locations in the Southern North Sea region around the Wadden Sea (from 2 to 10 °E and from 51 to 55 °N) and for the Peruvian continental margin (from 10 to 15 °S). Assemblage data for the North Sea region include 135 stations from the literature[51–57] and one at Friedrichskoog (from this study). Assemblage data for Peru include 35 stations[1,29].

The $\sum DIP_{i\,sed.}$ for 14 stations off Peru has previously been calculated, and is derived directly from ref. 1 For the remaining stations, $\sum DIP_{i\,sed.}$ was calculated according to equation (2) using the composition of benthic foraminiferal assemblages and intracellular phosphate content for each species (phosphate$_{in}$; Table 1):

$$\sum DIP_{i\,sed.} = \sum A_n \times phosphate_{i\,n} \times 10^{-9}, \qquad (2)$$

where $A_n$ is the abundance (living) of foraminiferal species $n$ (in ind m$^{-2}$) and phosphate$_{in}$ is the mean intracellular phosphate content of species $n$ (in pmol ind$^{-1}$). Individual phosphate storage data for *A. veneta* were not used for these calculations, because it was the only species for which no measurements were available from environmental samples (from laboratory cultures only). For species with unknown intracellular phosphate storage that share a genus with other species for which intracellular phosphate storage had already been determined, the average individual intracellular phosphate storage for this genus was used. Other species with unknown phosphate storage were excluded from the calculations. Finally, mmol m$^{-2}$ was converted to g m$^{-2}$ using the molar mass of phosphate (approximately 95 g mol$^{-1}$). All assemblage data used for caclulations are summarized in Supplementary Table ST8 (Southern North Sea) and Supplementary Table ST9 (Peru) as downloadable spreadsheets.

### Estimation of coastal riverine P runoff

Riverine phosphate runoff for the most important river estuaries of the Southern North Sea (Rhine, Meuse, Noordzeekanaal, Ijsselmeer, Ems, Weser and Elbe) was recorded for the year 2019 from the report of a monitoring programme[58]. Riverine total P runoff to the Peruvian coast from 10 to 15 °S, was taken from a global modelling study for the year 2015[59].

### Reporting summary

Further information on research design is available in the Nature Portfolio Reporting Summary linked to this article.

### Data availability

Publicly available protein sequences and transcriptomes were downloaded from the NCBI database (https://www.ncbi.nlm.nih.gov/) using the following accessions: GIDR00000000.1 (*A. confertitesta*), GIHI00000000.1 (*G. pacifica*) and GCA_000512085.1 (*R. filosa*). Raw data for the transcriptome assembly of *A. veneta* were obtained from the Sequence Read Archive (SRR18700766). Accessions (NCBI and KEGG databases) for individual creatine kinase sequences used are included in Supplementary Information. All sequence data in metabarcoding results (SRR1300434 and MK032924) are also available from the NCBI database (above). All other data from this study are available in the main text or Supplementary Information. Maps were created using Ocean Data View software (https://odv.awi.de/).

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

**Acknowledgements** We thank the scientific party and crew of RV *Meteor* cruise no. M176/2, chief scientist E. Achterberg, captain, crew and scientists on RV *Yokosuka*, the operation teams on HOV *Shinkai 6500* of JAMSTEC and the crew of the dive vessel *EastCom* for their support at sea. N.G. thanks J. Schönfeld and J. Wollenburg for helpful support with *Elphidium* taxonomy. The study is a contribution to the Centre for Earth System Research and Sustainability of Universität Hamburg. Funding was provided by Deutsche Forschungsgemeinschaft, through Heisenberg grant nos. GL 999/3-1 and GL 999/4-1 to N.G. Additional funding was provided by the Ocean Frontiers Institute to N.G. through the Visiting Fellowship Programme (2020). H.N. and J.R. received funding from the Japan Society for the Promotion of Science (grant nos. 21H01202 to H.N. and 22KF0424 to J.R. and H.N).

**Author contributions** Experimental strategy and study design were conceptualized by N.G. and H.N. Methodological work was executed by N.G., J.R., C.W., Y.I., T.H., A. Mutzberg, A. Makabe, S.O., A.T. and H.N. Sample preparation for, and measurements with, cryo-SEM−EDS and TEM−EDS were performed by S.O., J.R., A.T., H.N. and N.G. NMR measurements were performed by T.H. Comparative genomics were carried out by C.W. and Y.I. Sampling was the responsibility of N.G., J.R., C.A., M.A., D.E., H.F., A.G.M., R.H.-H., Y.M., S.N., S.R., G.S., Z.S., Z.Z. and H.N. Funding was acquired by N.G., G.S., H.N., J.R., M.K. and C.A. The main manuscript draft was written by N.G. All authors contributed to writing and editing of the manuscript.

**Funding** Open access funding provided by Universität Hamburg.

**Competing interests** The authors declare no competing interests.

**Additional information**
**Correspondence and requests for materials** should be addressed to Nicolaas Glock.

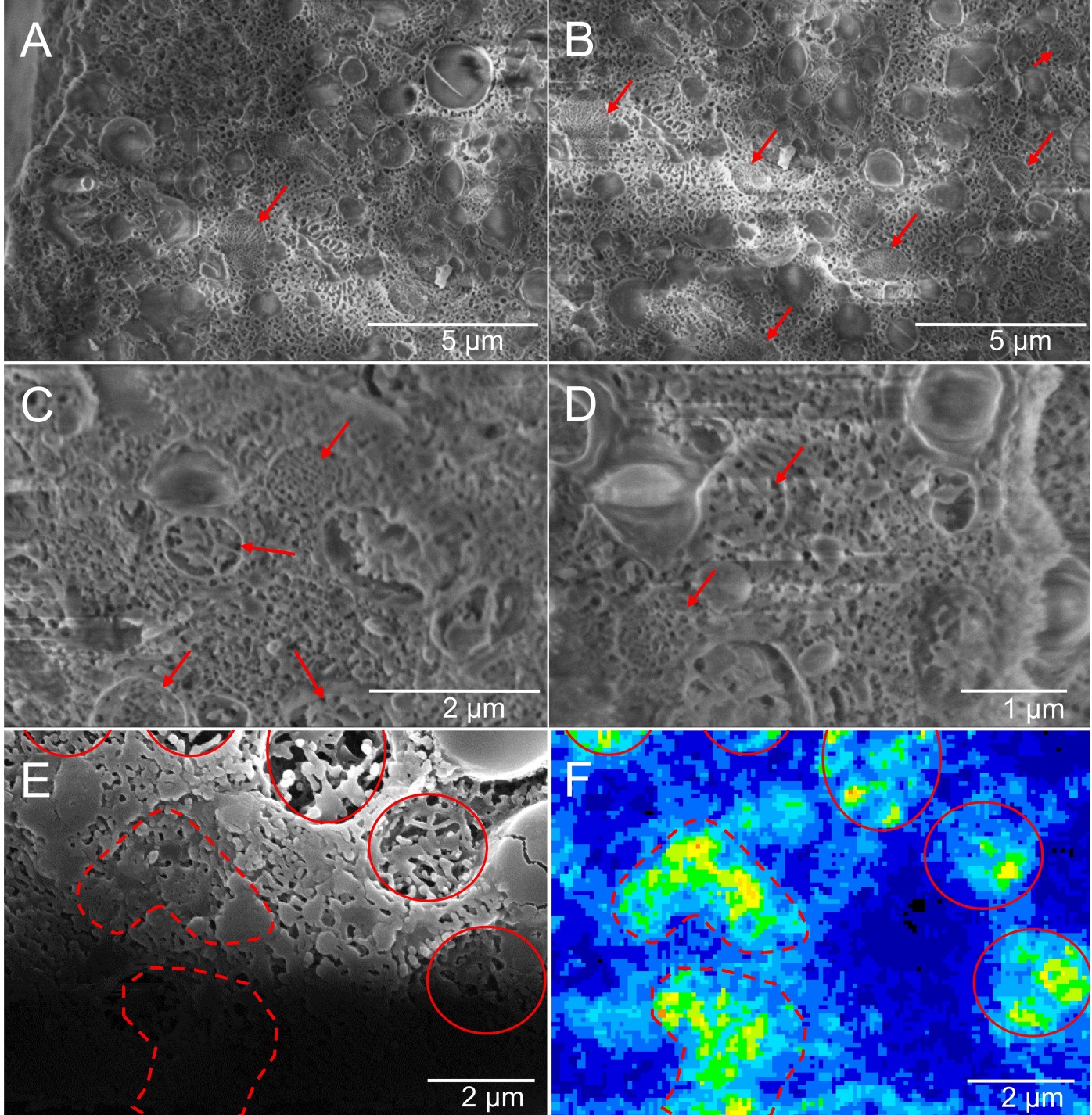

**Extended Data Fig. 1 | Ultrastructure of *Ammonia veneta*.** A-D: Supplementary cryo-SEM secondary electron images on a cross-sections of cryo-fractured *A. venata* specimens. The abundant circular structures marked with red arrows that are presumably acidocalcisomes. E&F: Cryo-SEM image (E) and P-distribution, mapped with Cryo-EDS (F), of the same region. These are the same images, as shown in Fig. 3b but shown individually for a better comparison. Red circles mark P accumulations in structures that ae presumably acidocalcisomes. Area cornered by red dashed lines show P-accumulations, which cannot be assigned to any visible structures on the cryo-SEM image. Possibly these accumulations are located slightly below the sample surface. The the cryo-SEM/EDS experiment has been repeated on 8 different specimens of *A. veneta*.

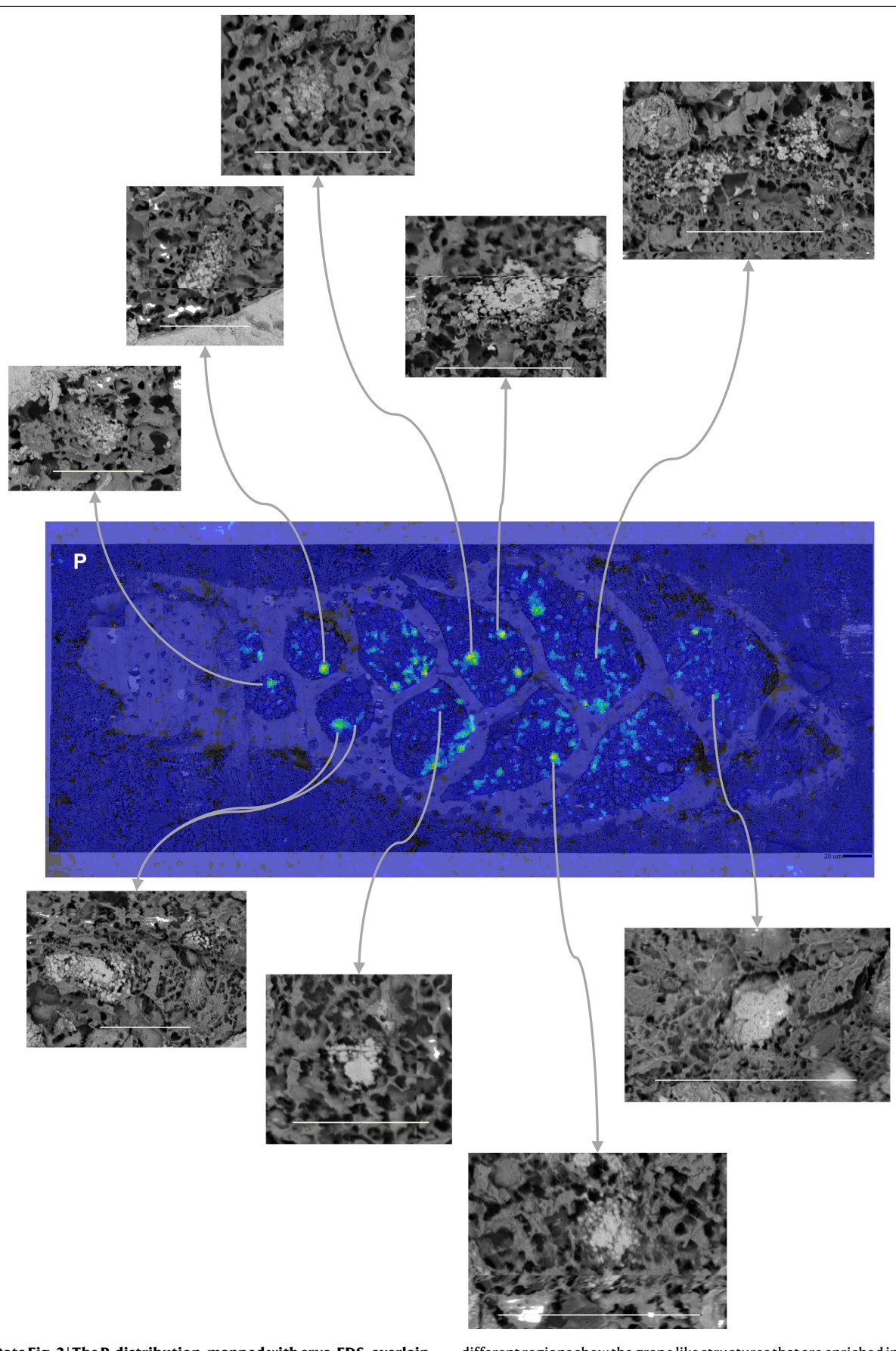

**Extended Data Fig. 2 | The P-distribution, mapped with cryo-EDS, overlain by the cryo-SEM image of the same area on a cryo-fractured *Bolivina spissa* specimen.** (Same specimen as shown in Fig. 2). The cryo-SEM close ups of the different regions show the grape like structures that are enriched in P, Ca and Mg. All white scale bars are 10 µm. The the cryo-SEM/EDS experiment has been repeated on 3 different specimens of *B. spissa*.

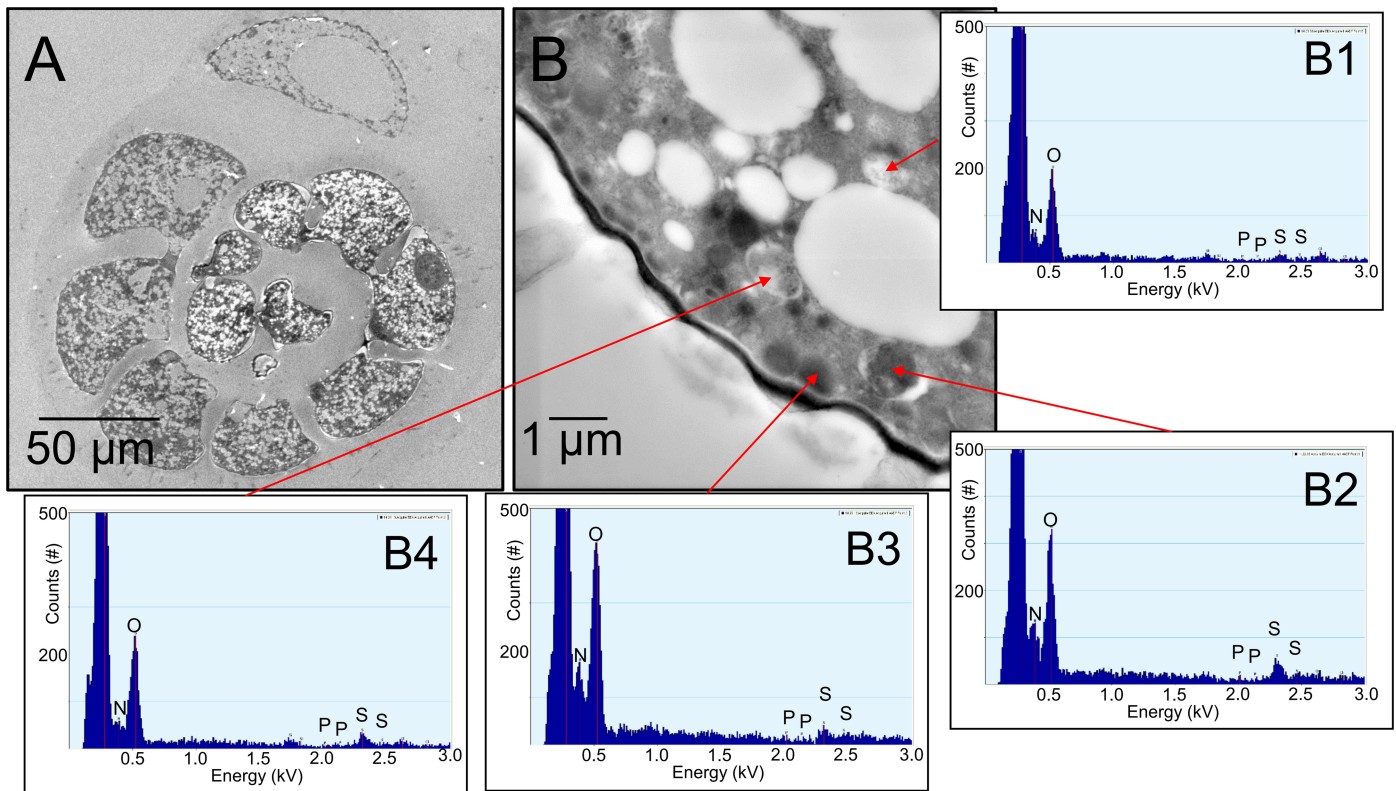

**Extended Data Fig. 3 | Ultrastructure and EDS spectra of *Ammonia veneta*.** TEM images of *A. venata* thin-sections (A&B) and locations and related spectra of EDS point measurements (B1-B4). Note the abundant empty vesicles with 0.5–3 µm diameter and the absence of phosphorus in EDS spectra. The TEM/EDS experiment of 3 different specimens of *A. veneta*.

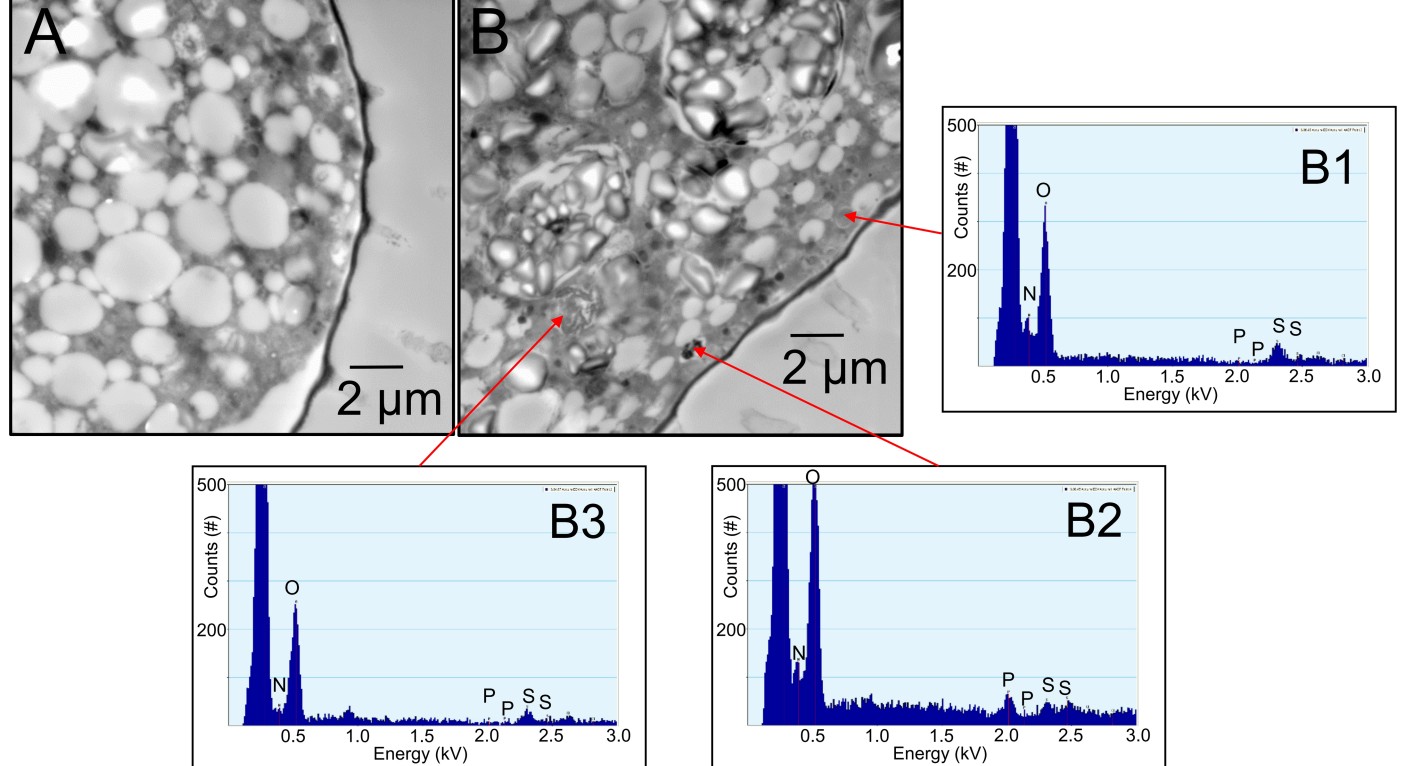

**Extended Data Fig. 4 | Ultrastructure and EDS spectra of *Ammonia veneta*.** TEM images of *A. venata* thin-sections (A&B) and locations and related spectra of EDS point measurements (B1-B3). Note the abundant empty vesicles with 0.5–3 μm diameter and the absence of phosphorus in EDS spectra B1 & B3. Only the vesicle, which still had visible content showed a visible phosphorus peak in the EDS spectrum (B2). The TEM/EDS experiment of 3 different specimens of *A. veneta*.

| | |
|---|---|

# Reporting Summary

## Statistics

For all statistical analyses, confirm that the following items are present in the figure legend, table legend, main text, or Methods section.

| n/a | Confirmed | |
|---|---|---|
| ☐ | ☒ | The exact sample size ($n$) for each experimental group/condition, given as a discrete number and unit of measurement |
| ☐ | ☒ | A statement on whether measurements were taken from distinct samples or whether the same sample was measured repeatedly |
| ☒ | ☐ | The statistical test(s) used AND whether they are one- or two-sided<br>*Only common tests should be described solely by name; describe more complex techniques in the Methods section.* |
| ☒ | ☐ | A description of all covariates tested |
| ☒ | ☐ | A description of any assumptions or corrections, such as tests of normality and adjustment for multiple comparisons |
| ☐ | ☒ | A full description of the statistical parameters including central tendency (e.g. means) or other basic estimates (e.g. regression coefficient) AND variation (e.g. standard deviation) or associated estimates of uncertainty (e.g. confidence intervals) |
| ☒ | ☐ | For null hypothesis testing, the test statistic (e.g. $F$, $t$, $r$) with confidence intervals, effect sizes, degrees of freedom and $P$ value noted<br>*Give P values as exact values whenever suitable.* |
| ☒ | ☐ | For Bayesian analysis, information on the choice of priors and Markov chain Monte Carlo settings |
| ☒ | ☐ | For hierarchical and complex designs, identification of the appropriate level for tests and full reporting of outcomes |
| ☒ | ☐ | Estimates of effect sizes (e.g. Cohen's $d$, Pearson's $r$), indicating how they were calculated |

*Our web collection on statistics for biologists contains articles on many of the points above.*

## Software and code

Policy information about availability of computer code

| Data collection | NMR spectroscopy: Data was acquired, using TopSpin version 3.6.4 and all all spectra were processed utilizing Topspin Version 4.1.4.<br>cryo-SEM/EDS: EDS data was acquired using TeamEDS, version 4.6.0052.0238.<br>Geographic maps were created using Ocean Data View (ODV) version 5.5.1.<br>Protein homologs in publicly available protein sequences were identified using the KEGG KAAS tool (https://doi.org/10.1093/nar/gkm321)<br>Raw reads for metabarcoding were quality-filtered with the FASTX-Toolkit 0.0.13<br>For the transcriptome assembly for Ammonia veneta the Trinity assembly tool (Version 2.15.1) was used. |
|---|---|
| Data analysis | NMR spectroscopy: Data was acquired, using TopSpin version 3.6.4 and all all spectra were processed utilizing Topspin Version 4.1.4.<br>cryo-SEM/EDS: EDS data was acquired using TeamEDS, version 4.6.0052.0238.<br>Geographic maps were created using Ocean Data View (ODV) version 5.5.1.<br>Protein homologs in publicly available protein sequences were identified using the KEGG KAAS tool (https://doi.org/10.1093/nar/gkm321)<br>Raw reads for metabarcoding were quality-filtered with the FASTX-Toolkit 0.0.13<br>For the transcriptome assembly for Ammonia veneta the Trinity assembly tool (Version 2.15.1) was used. |

For manuscripts utilizing custom algorithms or software that are central to the research but not yet described in published literature, software must be made available to editors and reviewers. We strongly encourage code deposition in a community repository (e.g. GitHub). See the Nature Portfolio guidelines for submitting code & software for further information.

## Data

Policy information about availability of data

All manuscripts must include a data availability statement. This statement should provide the following information, where applicable:

- Accession codes, unique identifiers, or web links for publicly available datasets
- A description of any restrictions on data availability
- For clinical datasets or third party data, please ensure that the statement adheres to our policy

Publicly available protein sequences and transcriptomes were downloaded from NCBI database (https://www.ncbi.nlm.nih.gov/) via following accessions: GIDR00000000.1 (Ammonia confertitesta) and GIHI00000000.1 (Globobulimina pacifica) and GCA_000512085.1 (Reticulomyxa filosa). Raw data for the transcriptome assembly of Ammonia veneta was obtained from the Sequence Read Archive (SRR18700766). Accessions (NCBI and KEGG databases) for the individual creatine kinase sequences used are included in the Supplementary information. All the sequence data in the Metabarcoting results section (SRR1300434 and MK032924) are also available in NCBI (https://www.ncbi.nlm.nih.gov/). All other data from this study are available in the main text or the supplementary materials.

## Research involving human participants, their data, or biological material

Policy information about studies with human participants or human data. See also policy information about sex, gender (identity/presentation), and sexual orientation and race, ethnicity and racism.

| | |
|---|---|
| Reporting on sex and gender | The terms "sex" and "gender" are not used in this study. |
| Reporting on race, ethnicity, or other socially relevant groupings | We are not reporting on "race, ethnicity, or other socially relevant groupings" in our study. |
| Population characteristics | This study was not involving human participants, their data, or biological material. |
| Recruitment | This study was not involving human participants, their data, or biological material. |
| Ethics oversight | This study was not involving human participants, their data, or biological material. |

Note that full information on the approval of the study protocol must also be provided in the manuscript.

# Field-specific reporting

Please select the one below that is the best fit for your research. If you are not sure, read the appropriate sections before making your selection.

☐ Life sciences ☐ Behavioural & social sciences ☒ Ecological, evolutionary & environmental sciences

For a reference copy of the document with all sections, see nature.com/documents/nr-reporting-summary-flat.pdf

# Ecological, evolutionary & environmental sciences study design

All studies must disclose on these points even when the disclosure is negative.

| | |
|---|---|
| Study description | The intracellular phosphate content of several species of benthic foraminifera from diverse marine environments has been quantified. In addition, we did ultrastructural analyses and element mapping on foraminiferal cells using cryoSEM/EDS and TEM/EDS to localize the intracellular phosphorous storage. For further characterization, in which form the phosphorous is stored, we performed extraction experiments and characterized the extracted phosphorous compounds using 31P-NMR. To characterize possible metabolic pathways, we performed comparative genomics on previously published transcriptomes and genomes of Ammonia veneta, Ammonia confertitesta, Globobulimina sp. and Reticulomyxa filosa. In addition, we used the results for the species specific phosphate storage and literature data about abundances of living foraminifera to estimate budgets for foraminiferal phosphate storage in the Southern North Sea and the Peruvian OMZ. Those were compared with the riverine phosphorous runoff in those regions. |
| Research sample | Most of the samples are freshly sampled benthic foraminiferal specimens, which have been directly sampled from seafloor sediments during several research cruises. Only the specimens of Ammonia veneta, that have been used for some of the TEM/EDS and cryo-SEM/EDS and intracellular phosphate concentration analyses are from a lab culture, which is described in detail in the methods section. As stated above, transcriptomes and genomes for the comparative genomics were already published before and have been taken from databases. All details are provided in the methods section. |
| Sampling strategy | We were specifically targeting different environments with various ranges of redox conditions. Specimens from the freshly collected samples have been chosen according to the foraminiferal species, that were present in the samples. Sediment samples have been processed immediately to provide results as close to the foraminifera´s natural habitat as possible. From these samples we picked as many living foraminiferal specimens as possible and tried to cover all foraminifera species that were present within the sample. We |

aimed to sample triplicates for all foraminifera species that were analyzed for the intracellular phosphate storage. For some species that were rare, we only were able to analyze one sample or duplicates, due to the lack of a sufficient number of specimens. All this is stated in datail within our datatables. Each sample contained between 1-70 living foraminifera specimens, depending on the average size and phosphate content of the analyzed foraminifera species. All samples were monospecific.
For the NMR analyzes, we needed huge samples, containing ~1000 living specimens. We chose the species Ammonia conferitesta for these analyses, since this species is very abundant within the easily accessible intertidal mudflats of Friedrichskoog. In addition, this species has the highest intracellular phosphate content, which reduced the required sample size.

| | |
|---|---|
| Data collection | Data about number of sampled specimens for intracelular phosphate quantification has been recorded by Nicolaas Glock. Data about the size of specimens has been acquired by measurements on microscope images and documented by Nicolaas Glock. Quantification of phosphate concentrations from the extracted samples have been measured using segmented flow injection analysis by Andre Mutzberg and Akiko Makabe who also did the data documentation. NMR-analysis and data documentation has been done by Thomas Hackl. TEM-EDS and cryo-SEM/EDS analyses and data documentation has been done by Satoshi Okada. |
| Timing and spatial scale | No timeline experiments have been performed for this study. |
| Data exclusions | No data was excluded from the analyses. |
| Reproducibility | As mentioned above: For the intracellular phosphate quantification, we aimed for triplicates for each foraminifera species. In some cases for rare speces, we only performed single analyses or duplicates. All this is documented in detail within our data tables. For the NMR-Data we used duplicates of samples, containing ~1000 specimens of Ammonia confertitesta and the spectra of both replicates were nearly the same as shown in the supplements. Another replicate of ~1500 specimens has been analyzed to measure the 31P-NMR spectrum directly within the living specimens. For TEM-EDS, we analyzed a total of 41 images on 3 different specimens of Ammonia venata. For the cryo-SEM/EDS we analyzed three different specimens of Bolivina spissa. All contained the P- and Ca-rich structures, described in the paper. |
| Randomization | Samples were divided by species level. Specimens from the newly collected samples have been chosen according to the foraminiferal species, that were present in the samples. From these samples we picked as many living foraminiferal specimens as possible and tried to cover all foraminifera species that were present within the sample. |
| Blinding | This study did not contain any randomized control trials, making blinding irrelevant. |

Did the study involve field work? ☒ Yes ☐ No

# Field work, collection and transport

| | |
|---|---|
| Field conditions | If possible, sampling time during ship cruises was chosen during good weather with low waves and winds to provide optimal conditions for sampling undisturbed sediments, using a multicorer. Weather and air temperature are not relevant for this study, except maybe at the intertidal mudflats of Friedrichskoog, since samples were retreived from the seafloor. Weather for sampling at the intertidal mudflats off Friedrichskoog was rainy in both November 2021 and May 2023. Air temperature in Friedrichskoog was 19.8°C in May 22nd 2023 and 6.8°C in November 25th 2021. |
| Location | All documented in table 2 of the main paper. |
| Access & import/export | No genetic material has been im- or exported for this study. No large animals have been sampled. The studied organisms include only foraminifera (microscopic protists).<br><br>Regarding sampling in the intertidal mudflats of Friedrichskoog during November 2021 and May 2023: Samples have been retrieved in compliance with state and national laws from a publically accesible non-protected spot in the German Wadden Sea at the town of Friedrichskoog. Sampling at Friedrichskoog was organized by the University of Hamburg.<br><br>Regarding to core retrievel during the RV Meteor cruise M176/2: Core retrieval was carried out exclusively outside the EEZ in compliance with international law and regulations of the neighbouring countries as mediated by the "Control Station German Research Vessels" of the University of Hamburg. Sampling cruise was organized by GEOMAR.<br><br>Samples during the Japanese cruises (R/V Kaimei cruise in September 2019, R/V Yokosuka cruise in May 2022, field trip to Hirakata Bay, Yokohama (Japan) in 2015) were collected from non-protected areas in Japanese territory or the Japanese EEZ in compliance with Japanese and international law. Sampling campaigns were organized by JAMSTEC.<br><br>Samples from the Bedford Basin (March 2022) were collected from non-protected areas in Canadian territory in compliance with Canadian and international law. The cruise was organized by Dalhousie University in Halifax. |
| Disturbance | The main disturbance by this study is related to the fact that foraminifera specimens had to be extracted from the sediments before the analyses. We aimed to minimize this disturbance by immediate sample processing and preparation after sampling. Only a few samples from the Bedford Basin had to be stored for 1-2 days in the cooling room, before it was possible to process them, due to the lack of time during the field trip. |

# Reporting for specific materials, systems and methods

We require information from authors about some types of materials, experimental systems and methods used in many studies. Here, indicate whether each material, system or method listed is relevant to your study. If you are not sure if a list item applies to your research, read the appropriate section before selecting a response.

## Materials & experimental systems

| n/a | Involved in the study |
|-----|----------------------|
| ☒ ☐ | Antibodies |
| ☒ ☐ | Eukaryotic cell lines |
| ☒ ☐ | Palaeontology and archaeology |
| ☒ ☐ | Animals and other organisms |
| ☒ ☐ | Clinical data |
| ☒ ☐ | Dual use research of concern |
| ☒ ☐ | Plants |

## Methods

| n/a | Involved in the study |
|-----|----------------------|
| ☒ ☐ | ChIP-seq |
| ☒ ☐ | Flow cytometry |
| ☒ ☐ | MRI-based neuroimaging |

## Plants

| Seed stocks | No plants were involved in this research. |
|-------------|-------------------------------------------|
| Novel plant genotypes | No plants were involved in this research. |
| Authentication | No plants were involved in this research. |

