## [Peer Review File · Nature]

Widespread occurrence and relevance of phosphate storage in foraminifera

Corresponding Author: Dr Nicolaas Glock

Version 0:

Reviewer comments:

Referee #1

(Remarks to the Author)

The manuscript of Glock et al describes the discovery of phosphate storage in benthic foraminifera in marine sediments. It was already reported a few years ago that foraminifera store phosphate anoxic marine settings (ref. 6), so this in itself is not a new discovery. However, what is new is the discovery that this phenomena is widespread (in at least 4 different locations now sampled) and , what is really new is the amount of P that is sequestered by the forams. For example, a single species in the Wadden Sea sequesters enough P to be a significant percentage of the P demand for Germany's fertilizer. This last point I think is very interesting and of high enough importance to justify consideration as a publication in Nature. However, in order for the manuscript to be published in Nature I think there are a few major changes that are required first.

Namely , there needs to be a main text figure showing a model or budget of marine P with the newly added component of benthic forams added. This should be done to drive the point home, how important this process is on a global scale - especially since the magnitude of the process is the new finding of the paper. In my opinion, the authors could really leverage this data set in this regard but currently its just some text and back of the envelope calculations that are hidden in the supplemental. This should be more present in the main text with a global budget/modelling analysis.

Secondly, there are numerous mistakes and inconsistencies between the supplemental figures and the text, and also the main text and what is shown in the figures. Many statements made in the text are inconsistent with what is shown in the figures and it appears that many wrong figures were uploaded to the supplemental by mistake and analyses are referred to that do not exist. I provide more extensive comments below.

Third, their justification of acidocalcisomes as the responsible vesicle needs alot more work. I could not follow their logic based on the text currently in the manuscript. I provide more extensive comments below.

If the authors can address each of these points I would be happy to review a revised version of the manuscript. I think if they are willing to make these changes it would be a very interesting paper and meet the high bar required for Nature.

Specific comments:

lines 32-33: This is the most significant part of the paper, I think for such a strong claim (which I agree appears to be supported by the authors data), a main text figure or model of some kind is needed. I provide some more suggestions below.

lines 72-75 : Were these sites chosen at random (it seems?) or was there some specific reason why these sites and not others? I understand the reason for comparing to the deep sea sites, but I wonder if there is a specific reason why those specific sites. Its ok if it was random, I think the sites you used are ok to make the claims in the paper, but readers like me will wonder whether there were additional science reasons why these sites.

line 85: Please provide more information on what about the size, shape and appearance make acidocalcisomes the most likely possibility, as opposed to other types of vacuoles.

line 93: I have searched the manuscript and supplement for an analysis of creatine kinase homologues but cannot find any. There is a figure legend in the supplement (S4) but figure S4 shows TEM and no genomic analysis. Can you please add the relevant figures to the manuscript.

lines 97-98: At this point, I am still wondering what specific evidence is showing that these are acidocalcisomes and not some other type of vesicle.

lines 118-120: Where are the O₂ values reported? I could not find them anywhere in the manuscript. Can you please add this information somewhere in a supplemental table for example. They are not in Fig 1 as the text currently implies.

lines 156-157: I am still not entirely convinced these are acidocalcisomes as the authors claim, is there any other evidence other than the size and shape? Why do you exclude the possibility that these are not potentially phagosomes or lysosomes, or any other kind of vacuole? Just using the size and shape seems a bit general to me, but perhaps phagosomes and lysosomes do not accumulate Ca and P? Can you please expand on your conclusion regarding the assignment of these as acidocalcisomes a bit more here. Moreover, Figure 3 A and B do not show location of the P in the vacuole structures - but this might be a mistake from the authors (See comments below).

lines 158-162: This is very confusing, you say that the acidocalcisome structures seen in cryo-SEM are "completely absent" and not observed in TEM yet Figure 3D clearly shows abundant empty round vesicles. Is this a typo or are you trying to say something else?

line 169: "...the absence of the round structures, observed with cryo-SEM but the presence of the many empty vesicles of similar size and shape in the TEM observations..." , this contradicts the photograph in Fig 3D which shows round vacuolar structures observed with TEM. How do you know the round structures seen in TEM (Fig 3D) and cryo-SEM (Fig 3A) are not the same?

line 172: The preceding text does not explain to me how you can conclude that the TEM vacuoles are acidocalcisomes.

lines 188-195: This is pretty speculative, I would remove it or reduce to one or two sentences.

lines 217-219: This paragraph does not explain why phosphate storage is an advantageous adaptation to rapid pH changes, or calcification. I would either revise or reduce this speculative paragraph.

lines 234-236: The cited study (reference 13) showed more than just that the forams encoded these genes, rather ref 13 showed that the creatine phosphate metabolism genes were expressed by forams in a natural anoxic sediment environment using metatranscriptomes. Please revise.

lines 236-238: It almost seems like the authors claim here that they are the first to show that these creatine metabolism genes are expressed, but this is not the case (see preceding comment). Moreover, supplemental Figure S4 does not show any analysis of creatine metabolism, it only shows a TEM analysis. Please correct this mistake. I think it was an error with the uploading of figures, since Fig S4 both have legends that are different than the figures that were uploaded.

lines 248 - 250: This same conclusion regarding rapid energetic bursts was already published in ref 13 (please update accordingly).

lines 266-270: Fig S1 needs to include some kind of standard measurements of the known P compounds that are claimed to have been detected. Otherwise, how can you know that these peaks are what you say they are?

line 270: You state "the peak of creatine phosphate" but no peak marked creatine phosphate is displayed in Fig S1.

line 272: By "diverse" do you actually refer to organic-P compounds?

lines 288-306: This section is the most interesting aspect of the whole manuscript, and should be greatly expanded and include a main text figure showing the global contribution of benthic forams to the P budgets (including the fertilizer use). There are biogeochemical modeling experts in the authors list, can they perhaps help to create a model of how the forams contribute to benthic P storage, and how this effects P limitation for primary productivity, and the scale of the storage as relates to fertilizer demand. I think this is a really high impact aspect of the authors study, but perhaps was an unexpected result but nevertheless I think this should be much more fleshed out and some other sections above reduced. The authors should create a new Figure 4 (replacing the existing one) that expands this analysis greatly. Perhaps you can come up with a model and some initial boundary conditions estimating the potential global storage of P by the forams. I think if the claim is made in the abstract, and is the high impact component, then a main text figure is required.

Comments on figures:

Figure 1: What is each individual point actually representing here? Is each point a different species of Forams? Or is each point the average of all forams from a single sample? Can you please provide the number of observations for each sample somewhere (n=), so that readers get a sense of how many cells (or samples?) are represented by the mean value for each

data point. Can you also please communicate in the legend whether each point represents mean values for each different species or a different sample set.

Figure 2: Can you confirm that the cytoplasm of *Bolivina* cells is present in all of the chambers during the active part of the cycle? My understanding was that most foram cytoplasm only is occupying the last most recently biocalcified chamber, closest to the aperture opening (to come out and attack the prey). If that were true then the phosphate inclusions seen in these images in the older chambers would be remnants of the activity and not from the current living cell (in the last most recent chamber).

Figure 3: The green colors in panel B don't match the location of the arrows in panel A. Is this a mistake? Since the location of the red arrows in A don't match the locations of the green colors in B. Also: the scale bars on panels A and C are different and it would be impossible to see any of the potential acidocalcisomes in Panel C at this magnification because they would only be 1 μm in size (or less). Can you provide a higher magnification of the TEM image in panel C to show that these structures are truly missing? How do you know that the vesicles in panel D are not the same vesicles seen in panel A as claimed in the text?

Figure S2: The legend says that panels E and F are the "same images" shown in Figure 3, but panel E in Figure S2 is obviously not the same image as that in Figure 3A. I think there is some kind of mistake here since the green regions in panel B of figure 3 don't match with the red arrows shown in panel A of figure 3.

figure S6: The legend of this tree says it is an 18S rRNA tree of ammonia, but in the method it says that Figure S6 is supposed to be showing a genomic analysis of creatine metabolism. Can you please correct these mistakes?

Figure 4: The creatine kinase energy circuit presented here was already published previously (ref 13) and then reviewed in a 2023 article by the first author. Therefore, I suggest to remove this since it is not really new. Rather, I suggest to replace Figure 4 with an additional figure providing initial boundary constraints on the role of phosphate storage in benthic forms for global P budgets and supply for fertilizer. And, or % of P flux to the ocean from land, are compensated for by the new PO_4 storage mechanism by the forams. Since the PO_4 storage in forams is not new, it is the magnitude of the process that is new, and this could make it a Nature publication. I think the authors need a figure like this, especially since they make the claim in the abstract about the 5% of Germany's P demand. There are some biogeochemical modeling experts in the author list, maybe they can come up with an initial model for constraining how the foram P storage mechanism plays a role in the global P budget?

Comments on Tables:

Table 1: This table should be moved to the supplemental in my opinion, and replaced with some of the more interesting aspects of the study like the global total amount of P stored by forams in sediments and how this relates to global P budgets (and fertilizer consumption). See comments above on how to improve Fig 4.

Referee #2

(Remarks to the Author)

The manuscript presents new findings on the widespread occurrence of intracellular phosphate storage in benthic foraminifera, building upon the first author's 2020 manuscript which suggested its possible role in adapting to low oxygen environments, observed in foraminifera from the Peruvian oxygen minimum zone. A notable contribution of this manuscript is its demonstration of this adaptation across various infaunal benthic foraminifera inhabiting diverse depth ranges (shallow and deep), thus expanding our understanding of its widespread prevalence. It is important to note that the majority, if not all, of the studied species belong to a single order Rotaliida, leaving open questions about the evolutionary extent of this adaptation within the foraminifera phylum.

Their results show that there is a considerable variation in phosphate concentrations among the studied species (with or without denitrification capabilities), with a potential correlation with oxygen-depleted environments. It is important to note that their proposition of phosphate storage as a crucial adaptation mechanism in foraminifera for coping with oxygen depletion heavily relies on self-citations from the 2020 paper. Furthermore it is worth noting that Orsi et al. (2020) have previously demonstrated the role of creatine phosphate as an ATP store in foraminifera, suggesting that it serves as an additional high-energy phosphate pool for cases of increased energy demand. Although I assume that this alone may not be enough as the sole metabolic mechanism under anoxic conditions.

The second novelty of this manuscript is that it shows that phosphate storage is maintained in acidocalcisomes, organelles rich in polyphosphate and other cations with diverse functions, which the authors claim as a first discovery in foraminifera. Indeed, their Cryo-SEM EDS analyses support the occurrence of phosphate storage within organelles resembling acidocalcisomes, making this an important discovery. They claim that the absence of these structures in TEM may be attributed to dehydration during sample preparation, prompting consideration for Cryo TEM for further validation (though not essential).

Another important contribution of this manuscript is the discussion on the metabolic function of phosphate storage, with

evidence presented for the expression of creatine kinase, a crucial gene in phosphate metabolism, in some of the examined species. However, it is important to highlight that the manuscript does not offer concrete information regarding the expression of this gene, only on its presence (line 237). A further exploration of its function could be, through qRT-PCR analyses comparing studied species with low and high levels of phosphate storage, or one of the high level species such *Ammonia confertitesta* under different oxygen concentrations.

Regarding specific comments, Figure 3 compares Cryo-SEM and TEM images of *Ammonia venata*, yet the phosphate concentration of this species is not indicated in Figure 1 or Table 1, leaving its phosphate storage status unclear. Additionally, Supplement s4 is erroneously labeled as 6 and requires correction.

Version 1:

Reviewer comments:

Referee #1

(Remarks to the Author)

I appreciate that the authors took my comments into consideration, especially the geospatial analysis of riverine P inputs (the new Figure 4) to the coastal area study sites with an estimation of how much P gets sequestered by the Foraminifera there. I really like how they put the amount of P sequestered in terms of number of days of runoff from the rivers, it puts it into perspective. This is a great addition to the study. The clarification of the inconsistencies between supplemental figures and supplemental notes are appreciated. I also appreciate the addition of the supplemental Table ST1 which provides useful background information on the O₂ and likely redox states. This is helpful for context. All of my other comments were basically addressed as well and at this point I only have a couple of additional comments for the authors to consider - after which I think its ready.

Fig 1: I thank the authors for their clarification of the nature of the samples, replicates here. But I still have a question, the update to the legend says that the N= is the number of samples. So, for example, the figure indicates that 3 samples were counted from the Peru OMZ in which *Bolivina costata* was present. However, different species have a different N= for the same location. This cannot make sense since an equal number of samples should have been counted for all species from the same location. Therefore, I wonder if when the authors write "samples" they actually are referring to individual foraminifera tests of a particular species that they saw in that particular sample. For example, is the figure actually showing that 3 tests of *B. costata* was observed for the Peru OMZ and 1 test of *B. spissa* was observed from Salami Bay? Can the authors please clarify this important point, whether N= is referring to number of samples or number of foram tests (it seems it is the later, which is different from what is currently in the legend. If that is the case, then it begs the question of how many samples were actually taken - was only a single sample looked at, and if so how much g of sediment seived to find the forams? Was an equal sampling effort applied to all samples? If the authors could clarify this I would be grateful.

Fig SF8 (the creatine kinase tree): Can you please add a few (maybe 10 addition sequences) more sequences from additional taxa (other eukaryotes, maybe mouse, chimp, or other protists for example) to this tree? Currently there are only 5 sequences which makes the bootstrap values at the nodes not very reliable in my opinion. In the authors rebuttal, they refer to Fig SF9 as the creatine kinase phylogeny, but this is an 18S rRNA gene tree. I think this was a mistake but I just want to check, in case.

Referee #2

(Remarks to the Author)

The manuscript introduces groundbreaking findings on the widespread occurrence of intracellular phosphate storage in benthic foraminifera, expanding on the first author's 2020 study, which proposed that this storage could aid adaptation to low-oxygen environments, particularly in the Peruvian oxygen minimum zone. A significant advancement in this manuscript is the confirmation that this adaptation occurs across various infaunal benthic foraminifera inhabiting diverse depth ranges, thus highlighting its broad prevalence.

I am very satisfied with the revised manuscript. I believe all points raised by the reviewers were adequately addressed. The accessibility to cutting-edge technologies promises to further illuminate this unique adaptation in future studies. As it stands, I am satisfied with the current version of the manuscript and excited to see further evidence of the remarkable biology of these magnificent organisms.

Version 2:

Reviewer comments:

Referee #1

(Remarks to the Author)

The authors have satisfactorily addressed all of my comments, and I recommend the manuscript for publication. I congratulate them on a job well done!

Rebuttal letter to the reviewers

We were really glad to receive positive feedback regarding our manuscript from both reviewers. The reviews were really constructive and we think our manuscript has significantly improved by following the suggestions of the reviewers. In the revised version of our manuscript, we considered all their suggestions. We added several new measurements, such as for example the measurement of the phosphate storage in *Ammonia veneta* from oxic and anoxic incubations, comparative transcriptomics of *Ammonia veneta*, several new supplementary figures, such as a phylogenetic tree of the creatine kinase in different foraminiferal species and detailed budget calculations of phosphate storage in the Southern North Sea and the Peruvian margin. Due to some of the new measurements, all authors agreed to add a new co-author, who provided important data (Akiko Makabe from JAMSTEC). We uploaded the revised files in two versions. One version without track changes at the beginning of the merged document and one version with track changes at the end of the merged document. This accounts for both the main manuscript and the supplementary information. Below, you can find a point by point response to the points of revisions.

Referees' comments:

Referee #1 (Remarks to the Author):

The manuscript of Glock et al describes the discovery of phosphate storage in benthic foraminifera in marine sediments. It was already reported a few years ago that foraminifera store phosphate anoxic marine settings (ref. 6), so this in itself is not a new discovery. However, what is new is the discovery that this phenomena is widespread (in at least 4 different locations now sampled) and, what is really new is the amount of P that is sequestered by the forams. For example, a single species in the Wadden Sea sequesters enough P to be a significant percentage of the P demand for Germany's fertilizer. This last point I think is very interesting and of high enough importance to justify consideration as a publication in Nature. However, in order for the manuscript to be published in Nature I think there are a few major changes that are required first.

Reply: We thank the reviewer for the positive and constructive feedback, regarding our manuscript. In our revised version of the manuscript, we added some new measurements and results of comparative genomics. In addition, as suggested by the reviewer, we extended the part about the relevance of foraminifera in phosphorus cycling considerably by adding two case studies. Below there is a point by point response to the specific comments and suggestions by the reviewer.

Referee #1: Namely, there needs to be a main text figure showing a model or budget of marine P with the newly added component of benthic forams added. This should be done to drive the point home, how important this process is on a global scale - especially since the magnitude of the process is the new finding of the paper. In my opinion, the authors could really leverage this data set in this regard but currently its just some text and back of the envelope calculations that are hidden in the supplemental. This should be more present in the main text with a global budget/modelling analysis.

Reply: As mentioned above, we added two case studies: One about the Southern North Sea and one about the Central Peruvian Margin. Those two regions were chosen to ensure that the budget calculation relies on substantial dataset and thus provide reliable estimations. In both cases, we gathered available data from the literature about living foraminifera assemblages and upscaled the total benthic foraminiferal phosphate storage, using the individual species specific phosphate storage

from our study and the living abundances of the different foraminifera species. The North Sea dataset includes 135 stations and the Peruvian dataset 35 stations. These were compared to the total riverine phosphate (North Sea) and total phosphorus (Peru) runoff to the coast. This new data indicates, that benthic foraminifera can buffer phosphorus runoff to the coast on timescales from weeks to months (details see in the responses to the more detailed comments below, in two new methods subsection, the new supplementary note SN8, a new figure 4 and some additional text in the main manuscript). Unfortunately a global budget analysis similar to our two case studies would be a bit out of scope of this study and would require datamining from all documented benthic foraminiferal assemblages worldwide, which is not realistic at this point. In addition, there would be many uncertainties that we cannot really account for. Especially, the data about P storage in deep sea species are very scarce and we only have data for a few species from the Mid-Atlantic-Ridge that really covers deeper water depths. The deep-sea is mainly dominated by soft-walled foraminifera and we lack any data about those taxa. About 84% of the ocean area is below 2000 m¹, which would bring huge uncertainties into a global budget estimation. A modelling study would be desirable for the future but the development of a model that integrates foraminiferal phosphate storage into a global model for marine phosphorus cycling would in our opinion require an own project that would likely take years of developing effort to withstand a proper review process.

Referee #1: Secondly, there are numerous mistakes and inconsistencies between the supplemental figures and the text, and also the main text and what is shown in the figures. Many statements made in the text are inconsistent with what is shown in the figures and it appears that many wrong figures were uploaded to the supplemental by mistake and analyses are referred to that do not exist. I provide more extensive comments below.

Reply: We apologize for this inconvenience. We realized that several of these “inconsistencies” are most likely related to the nomenclature within our supplement. Supplementary notes and supplementary figures often have similar numbers and were only differentiated by adding a “fig.” for supplementary figures (e.g. “see supplement fig.S2” for a figure instead of “see supplement S2” for a supplementary note). To avoid further confusion, we changed the nomenclature for supplementary notes and figures. Supplementary notes are now specifically cited as “supplementary note SN#”, while supplementary figures are cited as “supplementary fig. SF#”. Details regarding the specific points of critique by the reviewer, see below.

Referee #1: Third, their justification of acidocalcisomes as the responsible vesicle needs a lot more work. I could not follow their logic based on the text currently in the manuscript. I provide more extensive comments below.

Reply: We tried to adapt the text accordingly and formulated our discussion regarding the acidocalcisomes more carefully. See details at the responses to the more specific comments of the reviewer below.

Referee #1: If the authors can address each of these points I would be happy to review a revised version of the manuscript. I think if they are willing to make these changes it would be a very interesting paper and meet the high bar required for Nature.

Reply: We tried to follow the suggestions by the reviewer and adapted our manuscript accordingly. See below our point by point response to the specific comments.

Referee #1: Specific comments:

lines 32-33: This is the most significant part of the paper, I think for such a strong claim (which I agree appears to be supported by the authors data), a main text figure or model of some kind is needed. I provide some more suggestions below.

Reply: As mentioned above, we added some more detailed budget calculations for foraminiferal phosphate storage in sediments of the Southern North Sea (135 stations from literature) and the Peruvian OMZ (35 Stations from literature) and compared those to the riverine phosphorus runoff in these regions. Details about this are described in the revised manuscript and at one of the more detailed comments below. To address these results already in the abstract, we added the following sentence to the abstract:

“Budget calculations for the Southern North Sea and the Peruvian oxygen minimum zone indicate that benthic foraminifera may buffer riverine P runoff for ~37 days at the Southern North Sea and ~21 days at the Peruvian margin.”

Referee #1: lines 72-75 : Were these sites chosen at random (it seems?) or was there some specific reason why these sites and not others? I understand the reason for comparing to the deep sea sites, but I wonder if there is a specific reason why those specific sites. Its ok if it was random, I think the sites you used are ok to make the claims in the paper, but readers like me will wonder whether there were additional science reasons why these sites.

Reply: The main reason for the choice of the different sites was to cover a wide range of redox conditions and bathymetric ranges in the different environments. For example, very shallow intertidal mudflats with oxygen depleted pore waters, a seasonally hypoxic to anoxic shallow fjord basin, an oxygen minimum zone at intermediate water depths, intermediate to deep water hypoxia at Sagami Bay and the well-ventilated deep sea at the Mid-Atlantic-Ridge. However, the exact locations were partly “random”, due to the availability of the scientific cruises and samples. For example, the deep sea samples close to the Rainbow Vent Field could have been replaced with other locations from the Mid-Atlantic-Ridge but also provided the opportunity to study foraminiferal assemblages close to deep sea hydrothermal fields for further research. Due to the strong word limitation in Nature, we don't think it is necessary to discuss this in the paper.

Referee #1: line 85: Please provide more information on what about the size, shape and appearance make acidocalcisomes the most likely possibility, as opposed to other types of vacuoles.

Reply: We provide additional information about acidocalcisomes in literature to strengthen our arguments. In addition, we went again through our electron microscope images and did not find any of the vesicles we found to be larger than 2 μm , so we adapted the text accordingly. However, since we cannot finally prove, that the vesicles, we found are acidocalcisomes, we toned down our conclusions and refer to “possible” or “putative” acidocalcisomes. In line 85 we replaced “most likely acidocalcisomes” by “possibly acidocalcisomes”. The following passage has been added to the introduction to provide more information about size, shape and appearance of acidocalcisomes in literature, as the reviewer requested:

“In other organisms acidocalcisomes are usually enriched in P and Ca, as well, sometimes contain polyphosphate gels and granules and have a spherical shape². The size of acidocalcisomes varies depending on the organism. The typical diameter for protists is between 0.4 and 0.6 μm^2 but diameters $>1 \mu\text{m}$ are not uncommon³.”

Referee #1: line 93: I have searched the manuscript and supplement for an analysis of creatine kinase homologues but cannot find any. There is a figure legend in the supplement (S4) but figure S4 shows TEM and no genomic analysis. Can you please add the relevant figures to the manuscript.

Reply: We apology for this issue, which is most likely related to our choice on nomenclature, as I mentioned above. We did not refer to supplementary figure S4 but to supplementary note S4, which describes the results about the analysis of creatine kinase homologues. We did not have a figure, showing the phylogeny of the creatine kinase homologues and did not mean to refer to one. Nevertheless, to avoid further confusion, we not only adjusted the nomenclature of our supplement but we also added a figure that shows the phylogeny of the creatine kinase homologues with additional new results using the transcriptome of *Ammonia veneta*. The new figure is shown in the supplement (Fig.SF9).

Referee #1: lines 97-98: At this point, I am still wondering what specific evidence is showing that these are acidocalcisomes and not some other type of vesicle.

Reply: We rewrote this sentence and instead of “...storage of P in acidocalcisomes...” we wrote “...storage of P in organelles that are possibly acidocalcisomes...”.

Referee #1: lines 118-120: Where are the O₂ values reported? I could not find them anywhere in the manuscript. Can you please add this information somewhere in a supplemental table for example. They are not in Fig 1 as the text currently implies.

Reply: We did not report them in the original manuscript but added a supplementary table and a short paragraph in the revised version of the manuscript. This is actually not that trivial, since our sampling sites include locations with anoxic bottom and pore water, oxic bottom water but very shallow oxygen penetration depths into the pore waters (tidal mud flats) and locations with oxic bottom waters, where oxygen penetrates very deep (Mid Atlantic Ridge). Unfortunately, we do not have measured oxygen penetration depths in some environments and rely on literature data, that we cite in the revised manuscript. The following table is now cited in the main text and added to the supplement:

Tab.ST1: Bottom water O₂ concentrations and O₂ penetration depths in the sediments. At some locations, no literature data was available and we referred data from similar environments. *: Data has been taken from the Janssand intertidal area. **: O₂ pore water profiles have been taken in situ but in deeper water depths than the analysed foraminifera ($> 800 \text{ m}$). Bottom water O₂ concentrations were higher at the deeper stations and we expect the O₂ penetration depths to be even lower at the relevant shallower stations. ***: The O₂ penetration depths have been taken from another region at the Mid-Atlantic-Ridge in similar water depths with similar environmental conditions.

Location	Type	Bottom water O ₂ concentrations ($\mu\text{mol/kg}$)	O ₂ penetration depths (mm)
Sagami Bay (Japan)	Hypoxic	55 - 59 ⁴	2.6 – 17.8 ($\phi = 6.6 \pm 2.5$) ⁴
Friedrichskoog (intertidal mudflat)	Well ventilated bottom water with low O ₂ penetration	Saturated ^{5,*}	Low tide: 3 - 10 ^{5,*} High tide: 8 - 55 ^{5,*}

Bedford Basin	Seasonally hypoxic to anoxic	2 - 290 (depending on season) ⁶	~1 ⁶
Peruvian OMZ	Permanently suboxic to anoxic	1 - 13 (depending on water depth between 100 - 700 m) ⁷	< 4 ^{8,**}
Rainbow Vent Field (Mid-Atlantic-Ridge)	Well ventilated	242 - 246 between 2100 and 3100 m water depth ⁹	Up to 8000 (i.e. 8 m) ^{10,***}

Referee #1: lines 156-157: I am still not entirely convinced these are acidocalcisomes as the authors claim, is there any other evidence other than the size and shape? Why do you exclude the possibility that these are not potentially phagosomes or lysosomes, or any other kind of vacuole? Just using the size and shape seems a bit general to me, but perhaps phagosomes and lysosomes do not accumulate Ca and P? Can you please expand on your conclusion regarding the assignment of these as acidocalcisomes a bit more here.

Reply: The reviewer is right that we cannot completely exclude the possibility that those structures are other kinds of vacuoles. Especially autophagosomes have a similar size and shape and are known to include polyphosphates under stress as well³ and autophagy is known to be regulated by Ca. Nevertheless, we would be surprised, that the analyzed foraminifera would form so many autophagosomes, which usually would indicate a lot of stress, like for example phosphate starvation. In addition, why should foraminifera not have acidocalcisomes, organelles which are present in so many eukaryotes from protists to humans². In addition, we would exclude “normal” digestive vacuoles, since in other foraminifera they are typically much more irregular in size and shape, the interior looks very different and usually is not lost during TEM preparation. To assure, that we cannot prove that the structures, we found are acidocalcisomes, we toned down our interpretation and only discuss “putative acidocalcisomes”. Also, we expanded this part to acknowledge that we cannot exclude the presence of autophagosomes:

“The round structures, which are abundant within cells of *A. venata* that have been imaged with cryo-SEM (Fig. 3A&B) are similar in size and shape as acidocalcisomes³, organelles known to accumulate pyrophosphate (diphosphate) and granules that are enriched in Ca and polyphosphates and other metals such as Mg. Nevertheless, we cannot exclude that some of the structures are autophagosomes that can also accumulate polyphosphates³. Autophagosomes have a similar size and shape, although they usually show the presence of membranous debris and have a more irregular shape³. Digestive food vacuoles that are known from foraminifera have a completely different size and shape¹¹ and can most likely be excluded. The putative acidocalcisomes are either absent or empty in transmission electron microscope (TEM) images on thin sections of *A. venata*, since abundant and empty round vesicles of the same size are visible (Fig. 3C&D).”

Referee #1: Moreover, Figure 3 A and B do not show location of the P in the vacuole structures - but this might be a mistake from the authors (See comments below).

Reply: We apologize for this confusion. Our figure caption might have been a bit misleading. The images on Fig.3A&B are taken from different regions of a specimen of *A. veneta*. Fig.3B shows a layered image with the phosphorus distribution in the foreground and the cryo-SEM image in the background. The individual images (EDS mapping of phosphorus and the cryo-SEM image) of the layered image 3B can be found in the supplement (Fig.S2E&F). We adjusted the figure caption of Fig.3 and hope this clarifies this misunderstanding:

“Fig.3: Ultrastructure of *Ammonia venata* specimens. A: Cryo-SEM image (SE-mode) on a cross-section of a cryo-fractured specimen of *A. venata*. Note the abundant circular structures marked with red arrows that are possibly acidocalcisomes. B: The P-distribution, mapped with cryo-EDS, is shown in green on a cryo-fractured specimen of *A. venata*, overlain by the cryo-SEM image of the same region (SEM image and the P distribution are shown individually in Fig.S2). The circular, slightly P-enriched structures are absent on TEM images of thin sections of *A. venata* (C&D). Instead, there are abundant empty vesicles visible on the thin-sections, which indicates that these structures might have lost their content during fixation and subsequent embedding and polymerization processes. Only one of these vesicles has been found that was not empty (marked with the red arrow in D). This structure was the only structure with measurable P-content, using coupled TEM-EDS (see supplement fig.SF6).”

Referee #1: lines 158-162: This is very confusing, you say that the acidocalcisome structures seen in cryo-SEM are "completely absent" and not observed in TEM yet Figure 3D clearly shows abundant empty round vesicles. Is this a typo or are you trying to say something else?

Reply: This was most likely an unlucky formulation from our side. As the reviewer realized in one of the later comments, we hypothesize that the empty vesicles visible on TEM are actually the structures, visible on cryo-SEM but most likely lost their content during preparation. We adjusted the sentence accordingly:

“The putative acidocalcisomes are either absent or empty in transmission electron microscope (TEM) images on thin sections of *A. venata*, since abundant and empty round vesicles of the same size are visible (Fig. 3C&D).”

Referee #1: line 169: "...the absence of the round structures, observed with cryo-SEM but the presence of the many empty vesicles of similar size and shape in the TEM observations..." , this contradicts the photograph in Fig 3D which shows round vacuolar structures observed with TEM. How do you know the round structures seen in TEM (Fig 3D) and cryo-SEM (Fig 3A) are not the same?

Reply: See our response to the last comment. We reformulated this sentence accordingly:

“The presence of the many empty vesicles in the TEM observations, which are of similar size and shape as the round P-rich structures on the cryo-SEM/EDS images gives reason to speculate that many of the empty vesicles that were observed in *A. venata* using TEM are the same structures as the filled ones on the cryo-SEM observations and possibly acidocalcisomes.”

Referee #1: line 172: The preceding text does not explain to me how you can conclude that the TEM vacuoles are acidocalcisomes.

Reply: See response and reformulation of the sentence on the last two comments. We hope this clarifies, how we came to our hypothesis. As mentioned above, we cannot finally conclude about the acidocalcisomes and formulated everything a bit more carefully. Future studies might shed more light into the nature of the observed cell structures.

Referee #1: lines 188-195: This is pretty speculative, I would remove it or reduce to one or two sentences.

Reply: As suggested by the reviewer, we reduced this part to two sentences and clarified that parts of it are based on speculations:

“1. Fast osmoregulation: Rapid hydrolysis or synthesis of polyphosphates in acidocalcisomes, which increases or decreases the intracellular electrolyte concentration, has been shown as a reaction to hypo- or hyperosmotic stress in *Trypanosoma cruzi*, a protist belonging to Excavata¹². Both *A. confertitesta* and *B. costata* are often found in shallow marine environments that are strongly influenced by tidal cycles¹³⁻¹⁵ and thus experience drastic salinity changes, which lets us speculate, that this mechanism might be advantageous for rapid osmoregulation. “

Referee #1: lines 217-219: This paragraph does not explain why phosphate storage is an advantageous adaptation to rapid pH changes, or calcification. I would either revise or reduce this speculative paragraph.

Reply: We do not have any final proof for this mechanism but it is well documented, that the acidification of acidocalcisomes elevates the pH of other cell compartments and we definitely found elevated Ca concentrations in the structures, we observed in *B. spissa*. Why should both pH-regulation and Ca-storage not be advantageous for Calcification and environmental pH changes? Though, we agree that the hypothesis about the relevance of Ca-storage in acidocalcisomes as a Ca-reservoir for calcification was a bit too speculative, since there is also a lot of Ca in the surrounding pore- and bottom-water. Thus, we deleted this part of the paragraph but decided to keep the discussion about the intracellular pH regulation, which is one of the more important well documented functions of polyphosphate syntheses and hydrolysis. Though, we would agree to further reduce this paragraph, if the reviewer or the editor insists.

Referee #1: lines 234-236: The cited study (reference 13) showed more than just that the forams encoded these genes, rather ref 13 showed that the creatine phosphate metabolism genes were expressed by forams in a natural anoxic sediment environment using metatranscriptomes. Please revise.

Reply: Reformulated accordingly:

“Despite the relevance of phosphate storage in acidocalcisomes, the metatranscriptome of foraminifera from a natural anoxic sediment environment off Namibia revealed that they encode the genes for a creatine phosphate metabolism¹⁶.”

Referee #1: lines 236-238: It almost seems like the authors claim here that they are the first to show that these creatine metabolism genes are expressed, but this is not the case (see preceding comment).

Reply: We apologize for this inconvenience. It was not our intention to claim that we were the first to show that the creatine metabolism genes are expressed by foraminifera. We already cited reference 13 three times at this point and thought it would be clear, that this has been shown before. To address this issue, we reformulated the sentence accordingly:

“New comparative genomics and transcriptomics analyses on the species *A. confertitesta*, *A. veneta*, *G. pacifica* and *R. filosa* show, that these species also possess and/or express the enzyme creatine kinase and, thus, are able to synthesize and metabolize creatine phosphate (see supplementary note SN4 and fig.SF9).”

Referee #1: Moreover, supplemental Figure S4 does not show any analysis of creatine metabolism, it only shows a TEM analysis. Please correct this mistake. I think it was an error with the uploading of figures, since Fig S4 both have legends that are different than the figures that were uploaded.

Reply: As mentioned above, this misunderstanding was most likely related to the nomenclature in our supplement. We meant supplementary note 4, which discusses the comparative genomics results

in detail and we did not have a figure in the MS, that shows the phylogeny of the Creatine Kinase. This has been changed and we added an according figure (supplementary fig. SF9).

Referee #1: lines 248 - 250: This same conclusion regarding rapid energetic bursts was already published in ref 13 (please update accordingly).

Reply: Instead of citing reference 13 just at the end of the sentence, we cite it now at additional relevant positions of the sentence:

“Thus, it has been suggested before, that the creatine phosphate metabolism, observed in foraminifera from anoxic sediments at the Namibian Shelf¹⁶, provides an energy storage for sudden energetic bursts¹⁶, such as feeding by phagocytosis¹⁶, i.e., the vacuolar ingestion of food particles and prey¹⁶.”

Referee #1: lines 266-270: Fig S1 needs to include some kind of standard measurements of the known P compounds that are claimed to have been detected. Otherwise, how can you know that these peaks are what you say they are?

Reply: The reference standards we measured are Creatine Phosphate and (Sodium-)Pyrophosphate, because both compounds have very uncharacteristic singlet peaks with low NMR-shifts. We also added a reference spectrum of polyphosphate (Graham's Salt = Sodium Hexametaphosphate). We spiked the original samples, shown in fig.SF1, to obtain a matrix as close as possible to the original sample. The reference spectra of these three compounds are shown in a new fig.SF2. Other compounds were easier to identify, such as ATP, which shows its typical α -, β - and γ -peaks with their typical relative intensities.

Referee #1: line 270: You state "the peak of creatine phosphate" but no peak marked creatine phosphate is displayed in Fig S1.

Reply: The potential creatine phosphate peak was marked with an asterisk (*). This is also indicated in the figure captions: “The peak marked with “*” has a chemical shift very close to Creatine Phosphate.”

We hesitated to mark the peak as creatine phosphate directly in the figure, because we are not 100% sure, since the Creatine Phosphate standard had a slightly shifted peak. This might be related to a peak drift, because for the measurement of the reference standards we spiked the original samples. Unfortunately, this has been done a few days after the original spectrum of the sample has been measured. This is mainly related to the fact, that we were surprised by the outcome of the measurements and did not have the reference materials at hand. The peak of the Creatine Phosphate spike appeared about 0.77 ppm higher than during the original measurement of the sample. This could be related to drifts in the pH of the matrix or in temperature during the measurement. Hopefully future experiments will bring further clarification here. Nevertheless, we now marked the peak in the spectra of fig.SF1 as “Creatine Phosphate*” and clarified in the figure caption that the “*” indicates that this peak is “most likely” Creatine Phosphate. To address this issue in the revised MS, we extended the figure caption at the relevant section:

“The peak marked with “*” has a chemical shift very close to Creatine Phosphate (see fig.SF2) and most likely is Creatine Phosphate.”

Referee #1: line 272: By "diverse" do you actually refer to organic-P compounds?

Reply: No, we meant the possible metabolic uses that is related to the synthesis and hydrolysis of polyphosphates that we discussed above under the points 1-3 (energy storage for O₂ depletion, pH

and electrolyte regulation). The reviewer already stated that some of the points are hypotheses that could not finally be proven, so we toned this sentence down to "...the metabolic functions...might be quite diverse..." and to assure that we meant the functions mentioned above, we extended this sentence with examples:

"Although the metabolic functions of the intracellular phosphate storage in foraminifera might be quite diverse (e.g., energy supply under O₂ depletion, regulation of intracellular pH and electrolyte concentration...), it is easy to explain, why the deep-sea foraminifera from the MAR show lower intracellular phosphate accumulations."

Referee #1: lines 288-306: This section is the most interesting aspect of the whole manuscript, and should be greatly expanded and include a main text figure showing the global contribution of benthic forams to the P budgets (including the fertilizer use). There are biogeochemical modeling experts in the authors list, can they perhaps help to create a model of how the forams contribute to benthic P storage, and how this effects P limitation for primary productivity, and the scale of the storage as relates to fertilizer demand. I think this is a really high impact aspect of the authors study, but perhaps was an unexpected result but nevertheless I think this should be much more fleshed out and some other sections above reduced. The authors should create a new Figure 4 (replacing the existing one) that expands this analysis greatly. Perhaps you can come up with a model and some initial boundary conditions estimating the potential global storage of P by the forams. I think if the claim is made in the abstract, and is the high impact component, then a main text figure is required.

Reply: To address the reviewers constructive suggestion to add more detailed budget calculations to the paper, we created a new fig.4, that shows the distribution of foraminiferal phosphate storage in the Southern North Sea and the Peruvian coast between 10°S and 15°S. The calculations are based on the composition of living foraminiferal assemblages from 135 stations in the Southern North Sea and 35 Stations off Peru. These budgets are compared to the regional riverine phosphate and total phosphorus runoff from literature. More detailed budget calculations reveal that benthic foraminifera might buffer ~49 days of riverine phosphate runoff to the Southern North Sea and ~25 days of riverine total phosphate runoff to the Peruvian coast between 10°S and 15°S (see supplementary note SN8). Unfortunately, it would be impossible to provide such calculations globally, due to the huge amount of published compositions of living foraminiferal assemblages out there but we think that these estimates already show quite clearly the importance that foraminifera have in phosphorus cycling also on a global scale, since they are so important in two so different environments. More detailed future studies might be able to expand these estimates to a global scale. Regarding these new calculations, we provided not only a new figure but also two new sections in the methods chapter ("Calculation of total phosphate storage in living foraminiferal assemblages" and "Estimation of coastal riverine phosphorus runoff"), a new supplementary note SN8 and a new paragraph in the main manuscript:

"A more detailed analysis for benthic foraminiferal phosphate storage in sediments from 135 stations in the Southern North Sea and 35 stations at the Peruvian Margin reveals that benthic foraminifera might be an effective buffer for riverine phosphorus runoff in these regions (see fig.4 and supplementary note SN8). The total foraminiferal phosphate storage in the region of interest of the Southern North Sea ($0.0059 \pm 0.0014 \text{ g m}^{-2}$, 1SE) equals ~37 days of riverine phosphate runoff in this region (2583 t/yr). The foraminiferal assemblages between 10°S and 15°S off Peru store $0.0315 \pm 0.0101 \text{ g m}^{-2}$ (1SE) and thus may buffer ~21 days of riverine total phosphorus runoff in this region. Note that total riverine "phosphate" runoff from this region was not available and the budgets had to be calculated slightly different by using the riverine total "phosphorus" runoff (see supplementary note SN8).

At the Peruvian OMZ the foraminiferal phosphate storage might be of special importance. Phosphate from remineralized organic matter adsorbs to iron-oxide layers under oxic conditions and is effectively trapped in oxic sediments, while it escapes to the water column and is efficiently recycled under strongly O₂ depleted conditions⁵⁹. The high abundance of benthic foraminifera in this region might dampen this phosphate recycling, curve down productivity, and thus act as a negative feedback mechanism to the expansion of the OMZ that has been observed since the 1960s⁶⁰.”

And an additional sentence in the last paragraph of the main manuscript:

“The budget calculations about the total phosphate stored in benthic foraminifera in the Southern North Sea and the Peruvian OMZ indicate that this phenomenon is also globally important.”

Referee #1: Comments on figures:

Figure 1: What is each individual point actually representing here? Is each point a different species of Foram? Or is each point the average of all forams from a single sample? Can you please provide the number of observations for each sample somewhere (n=), so that readers get a sense of how many cells (or samples?) are represented by the mean value for each data point. Can you also please communicate in the legend whether each point represents mean values for each different species or a different sample set.

Reply: This is a very good point! In Fig. 1a each column represents the average for each species. In the revised figure, we provided the number of samples that have been analyzed for each species above the columns and added all the data of all individual samples as grey crosses. In addition, we added the following to the figure caption:

““n” is the number of samples that have been measured for each species. Grey crosses indicate data of individual samples.”

In fig. 1b each datapoint represents a single sample and is not ordered by species. We thought about putting the number of foraminiferal individuals in each sample next to the datapoint, but that would have made the figure way too busy. So we decided to refer to supplementary table ST2 for the number of individuals in each sample. In the revised paper we added the following part to the figure caption about 1b:

“Each datapoint represents one sample. Depending on the average body size of the species, each sample contained between 1 – 70 individuals (see supplementary tab. ST2).”

Referee #1: Figure 2: Can you confirm that the cytoplasm of *Bolivina* cells is present in all of the chambers during the active part of the cycle? My understanding was that most foram cytoplasm only is occupying the last most recently biocalcified chamber, closest to the aperture opening (to come out and attack the prey). If that were true then the phosphate inclusions seen in these images in the older chambers would be remnants of the activity and not from the current living cell (in the last most recent chamber).

Reply: The cytoplasm of foraminifera occupies the whole test and can be found even within the proloculus. Three dimensional observations of foraminiferal cytoplasm show that not only cytoplasm but also vacuolized seawater can be found in the older parts of the test (Nomaki et al., 2015¹⁷). Also organelles like mitochondria can be found in older parts of the test, often clustered behind pores and even nuclei have been observed within the proloculus. Actually, the last (newest) chamber is often not completely occupied with cytoplasm. Therefore phosphate inclusions seen in these images in the

older chambers are also play a role in their metabolism.

Referee #1: Figure 3: The green colors in panel B don't match the location of the arrows in panel A. Is this a mistake? Since the location of the red arrows in A don't match the locations of the green colors in B.

Reply: See our response above. A&B are not on the same region within the specimen. Figure 3B is a layered image that shows the phosphorus distribution from the EDS mapping in the foreground and the cryo-SEM image in the background. The individual images (EDS mapping of phosphorus and the cryo-SEM image) of the layered image 3B can be found in the supplement (Fig.S2E&F). We adjusted the figure caption of Fig.3 and hope this clarifies this misunderstanding (see our response in the related comment above).

Referee #1: Also: the scale bars on panels A and C are different and it would be impossible to see any of the potential acidocalcisomes in Panel C at this magnification because they would only be 1 μm in size (or less). Can you provide a higher magnification of the TEM image in panel C to show that these structures are truly missing? How do you know that the vesicles in panel D are not the same vesicles seen in panel A as claimed in the text?

Reply: This point has already been addressed in a few comments above. We actually intended to argue that some of the empty vesicles on TEM are the structures, shown in fig.3 A and might actually be acidocalcisomes that lost their content during TEM preparation. Due to some unlucky formulations from our side, we gave the reviewer the impression of the opposite. We adapted the text in the main manuscript and the figure caption accordingly and hope that this is much clearer within the revised manuscript. The relevant text changes can be found in the responses to some of the comments above. Related to the magnification issue of Fig.3C: We turned panels C&D by 90° which allowed us to magnify them a little bit without creating a new figure. Further magnification wouldn't reveal any more details, due to the resolution of the image. Panel D is a magnified spot on the upper left part of the whole cell image in Panel C.

Referee #1: Figure S2: The legend says that panels E and F are the "same images" shown in Figure 3, but panel E in Figure S2 is obviously not the same image as that in Figure 3A. I think there is some kind of mistake here since the green regions in panel B of figure 3 don't match with the red arrows shown in panel A of figure 3.

Reply: See our responses above. Figure 3 A&B are not on the same region within the specimen. Figure 3B is a layered image that shows the phosphorus distribution from the EDS mapping in the foreground and the cryo-SEM image in the background. The individual images (EDS mapping of phosphorus and the cryo-SEM image) of the layered image 3B can be found in the supplement (Fig.S2E&F). We adjusted the figure caption of Fig.3 and hope this clarifies this misunderstanding (see our response in the related comments above).

Referee #1: figure S6: The legend of this tree says it is an 18S rRNA tree of ammonia, but in the method it says that Figure S6 is supposed to be showing a genomic analysis of creatine metabolism. Can you please correct these mistakes?

Reply: As the caption of supplementary note S4 indicated, this note describes the results about the comparative genomics, but also of the metabarcoding analysis. The second paragraph, where figure S6 was cited, refers to the 18S rRNA metabarcoding. Most likely it was confusing, that we did not divide the two paragraphs by subheadings. In the revised version of the manuscript, we added subheadings to supplementary note S4. Subheading on the first paragraph now is "Comparative

genomics:” and of the second paragraph “Metabarcoding results:”.

Referee #1: Figure 4: The creatine kinase energy circuit presented here was already published previously (ref 13) and then reviewed in a 2023 article by the first author. Therefore, I suggest to remove this since it is not really new. Rather, I suggest to replace Figure 4 with an additional figure providing initial boundary constraints on the role of phosphate storage in benthic forms for global P budgets and supply for fertilizer. And, or % of P flux to the ocean from land, are compensated for by the new PO₄ storage mechanism by the forams. Since the PO₄ storage in forams is not new, it is the magnitude of the process that is new, and this could make it a Nature publication. I think the authors need a figure like this, especially since they make the claim in the abstract about the 5% of Germany's P demand. There are some biogeochemical modeling experts in the author list, maybe they can come up with an initial model for constraining how the foram P storage mechanism plays a role in the global P budget?

Reply: As the reviewer suggested, we deleted fig.4 A that shows the creatine kinase energy circuit. We decided to move the rest of the original figure to the supplement, since it might provide valuable information for readers that are not that familiar with the metabolic effects of polyphosphate hydrolysis and synthesis.

To address the reviewers constructive suggestion to add more detailed budget calculations to the paper, we created a new fig.4, that shows the distribution of foraminiferal phosphate storage in the Southern North Sea and the Peruvian coast between 10°S and 15°S. Details about this have been already given in the response to one of the comments above.

Referee #1: Comments on Tables:

Table 1: This table should be moved to the supplemental in my opinion, and replaced with some of the more interesting aspects of the study like the global total amount of P stored by forams in sediments and how this relates to global P budgets (and fertilizer consumption). See comments above on how to improve Fig 4.

Reply: We exchanged figure 4 with a figure showing maps of foraminiferal phosphate storage at the Southern North Sea and the Peruvian OMZ in comparison with the regional riverine phosphate runoff in numbers. Though, we hesitate to move table 1 to the supplement. It might be, that the species specific phosphate storages are not that shiny as a table with a summary of the P-budgets but those are now well addressed in the additional text sections and the new figure 4. The species specific phosphate storages are a key result of this study and have been used to do the actual budget calculations. This might be of high interest for other biogeochemical modelers and researchers, focusing on foraminifera. If the reviewer insists, we would remove the table from the main manuscript but we think that it should not be hidden in the supplement.

Referee #2: (Remarks to the Author):

The manuscript presents new findings on the widespread occurrence of intracellular phosphate storage in benthic foraminifera, building upon the first author's 2020 manuscript which suggested its

possible role in adapting to low oxygen environments, observed in foraminifera from the Peruvian oxygen minimum zone. A notable contribution of this manuscript is its demonstration of this adaptation across various infaunal benthic foraminifera inhabiting diverse depth ranges (shallow and deep), thus expanding our understanding of its widespread prevalence. It is important to note that the majority, if not all, of the studied species belong to a single order Rotaliida, leaving open questions about the evolutionary extent of this adaptation within the foraminifera phylum.

Their results show that there is a considerable variation in phosphate concentrations among the studied species (with or without denitrification capabilities), with a potential correlation with oxygen-depleted environments. It is important to note that their proposition of phosphate storage as a crucial adaptation mechanism in foraminifera for coping with oxygen depletion heavily relies on self-citations from the 2020 paper. Furthermore it is worth noting that Orsi et al. (2020) have previously demonstrated the role of creatine phosphate as an ATP store in foraminifera, suggesting that it serves as an additional high-energy phosphate pool for cases of increased energy demand. Although I assume that this alone may not be enough as the sole metabolic mechanism under anoxic conditions.

Reply: We thank the reviewer for the constructive input, regarding our manuscript. The question regarding the evolutionary extend is a very interesting idea, although our dataset also includes some agglutinated and other foraminiferal taxa, which do not belong to the rotaliids: *Astrorhizida* (*R. algaeformis*), *Lituolida* (*L. crassimargo*), *Textulariida*, *Robertinida* (*H. elegans*), and *Buliminida*. We hope that we can provide a detailed analysis about the variations between the different orders of foraminifera in the future. Also, an analysis of planktic foraminifera and other rhizaria might be really interesting in the future. Of course, our study is mainly based on the Orsi et al., 2020 and Glock et al., 2020 papers, since they are the only studies so far published about phosphate storage and phosphorus metabolism in foraminifera. We are aware of some studies that have been done in the mid 1990s but they are only published as conference abstracts:

Langer, Martin R., West, O, Bernhard, J., Bowser, S. S. (1995): Intracellular "amorphous" calcium-phosphate: a new biomineral in foraminifera.. Geological Society of America Annual Meeting, New Orleans, LA. Abstracts with Program, Session 108, Band 27. Ausgabe No. 6 BTH 10, Seiten A-304

West, O., Bernhard, J.M., Bowser, S.S., Langer M.R. (1995) Intracellular refractile inclusions in benthic foraminifera 6th East-Coast Conference on Protozoa, June 2-3, Schenectady, NY, p. 24

Those found phosphorus and calcium rich structures, which most likely represent polyphosphates as well, in some allogromiids. The problem with citing these studies is that they have never been published. But it shows, that there are results of other studies that support our results.

Referee #2: The second novelty of this manuscript is that it shows that phosphate storage is maintained in acidocalcisomes, organelles rich in polyphosphate and other cations with diverse functions, which the authors claim as a first discovery in foraminifera. Indeed, their Cryo-SEM EDS analyses support the occurrence of phosphate storage within organelles resembling acidocalcisomes, making this an important discovery. They claim that the absence of these structures in TEM may be attributed to dehydration during sample preparation, prompting consideration for Cryo TEM for further validation (though not essential).

Reply: We appreciate that the reviewer recognizes and supports our interpretation about the acidocalcisomes. Reviewer #1 remarked that we also have to consider the possibility that those structures are other types of vesicles, such as autophagosomes. Thus, we toned our conclusions down a bit and extended out discussion regarding the acidocalcisomes. Unfortunately, we do not have access to Cryo-TEM for additional analyses but the methodology development of Cryo-

TEM/NanoSIMS is currently making big advances. So we hope that these advancements will reveal further details in the future.

Referee #2: Another important contribution of this manuscript is the discussion on the metabolic function of phosphate storage, with evidence presented for the expression of creatine kinase, a crucial gene in phosphate metabolism, in some of the examined species. However, it is important to highlight that the manuscript does not offer concrete information regarding the expression of this gene, only on its presence (line 237). A further exploration of its function could be, through qRT-PCR analyses comparing studied species with low and high levels of phosphate storage, or one of the high level species such *Ammonia confertitesta* under different oxygen concentrations.

Reply: Thanks a lot for those interesting suggestions for future analyses! For the revised manuscript, we actually added further comparative genomics analyses. We also tested *Ammonia veneta* for the presence of creatine kinase and provide a phylogenetic tree, that shows that the creatine kinase in foraminifera clusters close to the creatine kinase in muscle cells. We are also aware that some groups already culture *Ammonia confertitesta* under varying oxygen concentrations and hope that future collaborations will allow us to test different levels of gene expression between different species and different oxygen concentrations. A comparison between *Ammonia confertitesta* and *Ammonia veneta* might be interesting as well. As the reviewer suggested in the next comment, we added intracellular phosphate analysis for *Ammonia veneta* as well to our revised manuscript. It shows considerably lower intracellular phosphate content than *Ammonia confertitesta* but we have to state that the samples of *Ammonia veneta* were from lab cultures, while all other species we analysed were sampled in the natural environment. Therefore the P aggregations found from Cryo-SEM analyses of *Ammonia veneta* are thought to be reflecting high phosphate concentrations in some certain organelles.

Referee #2: Regarding specific comments, Figure 3 compares Cryo-SEM and TEM images of *Ammonia veneta*, yet the phosphate concentration of this species is not indicated in Figure 1 or Table 1, leaving its phosphate storage status unclear. Additionally, Supplement s4 is erroneously labeled as 6 and requires correction.

Reply: We did not have analyses of *Ammonia veneta*'s intracellular phosphate concentration before but we added the analyses and results for the revised manuscript. The specimens have been collected from lab cultures at Jamstec and included a sample of 75 specimens under anoxic conditions and a sample of 70 specimens under oxic conditions. The intracellular phosphate concentration in both samples was considerably lower than in *Ammonia confertitesta* from the field but comparable with other species within the lower third of the phosphate storage magnitude amongst species.

Regarding the labeling: This is most likely related to a problem regarding the nomenclature of our supplement. We did not create specific labels for supplementary notes and supplementary figures. The citation of supplementary note S4 refers to the supplementary note, where we describe the results of the comparative genomics and the metabarcoding. Supplementary figure S6 then is cited in supplementary note S4 and shows the phylogenetic tree of the metabarcoding analysis. This also led to some misunderstandings with reviewer #1 and we decided to change the whole nomenclature of our supplement. Supplementary notes are now referred to as "supplementary note SN#", supplementary figures are now "SF#" and supplementary tables are now "ST#".

Rebuttal letter to the reviewers

We thank both reviewers for the extremely positive feedback to our revised manuscript version. Reviewer #2 recommended to publish the paper as is and there were only two minor points of revisions by reviewer #1. We tried to address both points of revision in our newly revised manuscript and provide a point by point response below, what we have changed in the paper. In addition, we adapted the title of the paper to be in conformity with the limit of the title length in nature. The new title is as follows:

“Extensive phosphate storage in foraminifera is an adaptation to O₂ depletion”

Referees' comments:

Referee #1 (Remarks to the Author):

I appreciate that the authors took my comments into consideration, especially the geospatial analysis of riverine P inputs (the new Figure 4) to the coastal area study sites with an estimation of how much P gets sequestered by the Foraminifera there. I really like how they put the amount of P sequestered in terms of number of days of runoff from the rivers, it puts it into perspective. This is a great addition to the study. The clarification of the inconsistencies between supplemental figures and supplemental notes are appreciated. I also appreciate the addition of the supplemental Table ST1 which provides useful background information on the O₂ and likely redox states. This is helpful for context. All of my other comments were basically addressed as well and at this point I only have a couple of additional comments for the authors to consider - after which I think its ready.

Reply: We thank the reviewer for such a positive feedback and for appreciating our revisions! We tried to incorporate both requested changes in a revised version of the MS.

Referee #1: Fig 1: I thank the authors for their clarification of the nature of the samples, replicates here. But I still have a question, the update to the legend says that the N= is the number of samples. So, for example, the figure indicates that 3 samples were counted from the Peru OMZ in which *Bolivina costata* was present. However, different species have a different N= for the same location. This cannot make sense since an equal number of samples should have been counted for all species from the same location. Therefore, I wonder if when the authors write "samples" they actually are referring to individual foraminifera tests of a particular species that they saw in that particular sample. For example, is the figure actually showing that 3 tests of *B. costata* was observed for the Peru OMZ and 1 test of *B. spissa* was observed from Salami Bay? Can the authors please clarify this important point, whether N= is referring to number of samples or number of foram tests (it seems it is the later, which is different from what is currently in the legend. If that is the case, then it begs the question of how many samples were actually taken - was only a single sample looked at, and if so how much g of sediment sieved to find the forams? Was an equal sampling effort applied to all samples? If the authors could clarify this I would be grateful.

Reply: Nearly all individual samples contained multiple specimens of foraminifera. The number of individuals (1 – 75) was based on the size of the foraminiferal species. Only for rare and extremely large species, such as *Rhizammina algaeformis*, we used one single specimen for a single sample. The exact number of foraminiferal specimens for each sample can be found in supplementary tab. ST2. Of course the sampling effort depended on the abundances of living foraminifera in the sample. For example, in the mudflats, with population densities of several hundred living foraminifera per cm³ of

sediment it was relatively low effort to pick hundreds of living specimens (or even thousands for the NMR analyses). At the deep sea (Mid Atlantic Ridge) it was much harder to find enough living specimens and I ended up picking whole nights on the ship without enough specimens for a single sample. In those cases, we did not analyze the samples. We only analyzed samples where we also expected that the number of living individuals was high enough to be above detection limit of the method for the phosphate quantification. To clarify this issue, we added the following sentence to the figure caption of figure 1A:

“Each sample contained between 1 – 75 individuals depending on the size of the species (see supplementary tab. ST2). “

Referee #1: Fig SF8 (the creatine kinase tree): Can you please add a few (maybe 10 additional sequences) more sequences from additional taxa (other eukaryotes, maybe mouse, chimp, or other protists for example) to this tree? Currently there are only 5 sequences which makes the bootstrap values at the nodes not very reliable in my opinion. In the authors rebuttal, they refer to Fig SF9 as the creatine kinase phylogeny, but this is an 18S rRNA gene tree. I think this was a mistake but I just want to check, in case.

Reply: Yes, the reference to Fig SF9 in the rebuttal letter was wrong, because the figure numbers did change after we wrote this part of the letter and we forgot to update it. Fig. SF8 is the right one. As the reviewer requested now, we updated the phylogenetic tree in figure SF8 and instead of 5 taxa, it now includes 16. The following sentence was added to the according methods section:

“Additional creatine kinase sequences (annotated as K00933) were obtained from the KEGG database.”

In addition, we adapted the last sentence of the figure caption of Fig. SF8:

“The tree was rooted between the taxonomic groups Sar (+ *Guillardia theta*) and Opisthokonta.”

Referee #2 (Remarks to the Author):

The manuscript introduces groundbreaking findings on the widespread occurrence of intracellular phosphate storage in benthic foraminifera, expanding on the first author's 2020 study, which proposed that this storage could aid adaptation to low-oxygen environments, particularly in the Peruvian oxygen minimum zone. A significant advancement in this manuscript is the confirmation that this adaptation occurs across various infaunal benthic foraminifera inhabiting diverse depth ranges, thus highlighting its broad prevalence.

I am very satisfied with the revised manuscript. I believe all points raised by the reviewers were adequately addressed. The accessibility to cutting-edge technologies promises to further illuminate this unique adaptation in future studies. As it stands, I am satisfied with the current version of the manuscript and excited to see further evidence of the remarkable biology of these magnificent organisms.

Reply: We thank the reviewer for such a positive feedback and for appreciating our revisions!